# On retrieving sea ice freeboard from ICESat laser altimeter

Kirill Khvorostovsky[1], Pierre Rampal[1]

[1]Nansen Environmental and Remote Sensing Center, 5006 Bergen, Norway

*Correspondence to*: K. Khvorostovsky (kirill.khvorostovsky@nersc.no)

**Abstract.** Sea ice freeboard derived from satellite altimetry is the basis for estimation of sea ice thickness using the assumption of hydrostatic equilibrium. High accuracy of altimeter measurements and freeboard retrieval procedure are therefore required. As of today, two approaches for estimation of the freeboard using laser altimeter measurements from Ice, Cloud, and land Elevation Satellite (ICESat), referred to as tie-points (TP) and lowest-level elevation (LLE) methods, have been developed and applied in different studies. We reproduced these methods for the ICESat observation periods (2003-2008) in order to assess and analyze the sources of differences found in the retrieved freeboard and corresponding thickness estimates of the Arctic sea ice as produced by the Jet Propulsion Laboratory (JPL) and Goddard Space Flight Center (GSFC). Three main factors are found to affect the freeboard differences when applying these methods: A) the approach used for calculation of the local sea surface references in leads (TP or LLE methods), B) the along-track averaging scales used for this calculation, and C) the corrections for lead width relative to the ICESat footprint and for snow depth accumulated in refrozen leads. The LLE method with 100 km averaging scale, as used to produce the GSFC datasets, and the LLE method with a shorter averaging scale of 25 km both give larger freeboard estimates comparing to those derived by applying the TP method with 25-km averaging scale as used for the JPL product. These two factors, A) and B), contribute to the freeboard differences in approximately equal proportions, and their combined effect is, on average, about 6-7 cm. The effect of using different methods varies spatially: the LLE method tends to give lower freeboards (by up to 15 cm) over the thick multiyear ice and higher freeboards (by up to 10 cm) over first-year ice and thin part of multiyear ice; these higher freeboards dominate. We show that the freeboard under-estimation over most part of these thinner parts of sea ice can be reduced to less than 2 cm when using the improved TP method proposed in this paper. The corrections for snow depth in leads and lead width, C), are applied only for the JPL product and increase the freeboard estimates by about 7 cm, on average. Thus, different approaches to calculate sea surface references and different along-track averaging scales, from one side, and the freeboard corrections as applied when producing the JPL dataset, from the other side, are roughly compensating each other with respect to freeboard estimation. Therefore, one may conclude that the difference in the mean sea ice thickness between the JPL and GSFC datasets reported in previous studies should be attributed mostly to different parameters used in the freeboard-to-thickness conversion.

## 1 Introduction

The observed thinning of the Arctic sea ice cover during the last 15 years is one of the most sensitive indicators of the climate change (e.g. Stocker et al., 2013; Laxon et al., 2013). The main data source for retrieving the sea ice thickness over large-scale basins is the radar and laser satellite altimeter measurements of the sea ice freeboard, which is used to convert freeboard to thickness assuming the hydrostatic equilibrium of floating ice (e.g. Kwok et al., 2009; Laxon et al., 2003; Laxon et al, 2013; Ricker et al., 2014; Tilling et al., 2015). Using this particular conversion method, the uncertainty of the obtained sea ice thickness is equal to approximately ten times the one associated with the ice freeboard estimate. This stresses the need for very accurate altimeter measurements and freeboard retrieval procedure in order to minimize sea ice thickness uncertainty (e.g. Zygmuntovska et al., 2014), and increase the confidence level associated to the negative trend in Arctic sea ice volume reported in the last 2013 IPCC report (Vaughan et al., 2013).

In this paper we focus on uncertainty of total (snow plus sea ice) freeboard retrieval using laser altimeter measurements from Ice, Cloud, and land Elevation Satellite (ICESat). As compared to satellite radar altimetry, ICESat provides higher accuracy in elevation measurements over a comparatively smaller footprint of ~70 m, with a precision of about 2 cm (Kwok et al., 2004) and the single-shot accuracy of 13.8 cm (Zwally et al., 2002). A key step in freeboard estimation process is the determination of the local sea surface height that is used as reference elevation. The determination of the local sea surface height from geoid, modelled tides, and atmospheric pressure loading is rather uncertain. Therefore, a common method is to calculate a local reference elevation from ICESat measurements over open water (or thin ice) within leads. Several methods to detect such samples were proposed in a number of studies (Kwok et al., 2007; Zwally et al., 2008; Farrell et al., 2009).

The approach proposed by Kwok et al. (2007), referred as the tie-points (hereafter TP) method, is based on identification of sea surface reference points (tie-points), for which the deviation between the measured elevation and the local mean surface exceeds a given value. Indeed, they found a relationship between surface roughness and freeboard adjacent to new lead/crack openings associated with low reflectivity, and therefore used it for tie-points detection. Later, the TP method has been further developed by Kwok and Cunningham (2008) and Kwok et al. (2009), and applied by Kurtz et al. (2009) and Kurtz et al. (2011) to study sea ice thickness in the Arctic. A similar approach, based on the same roughness/freeboard relationship, has been defined and used by Markus et al. (2011) and Kurtz and Markus (2012) to retrieve freeboard of the Antarctic sea ice. However, one should note that the TP method has some limitations as it is based on an empirical relationship that may not be valid for some specific time and location.

Another approach, the so-called lowest-level elevation (hereafter LLE) method, was originally described and used in a study by Zwally et al. (2008) and later applied by, for example, Yi et al. (2011), Xie et al. (2013) or Kern and Spreen (2015) to retrieve freeboard of Antarctic sea ice, and by Yi and Zwally (2009) for the Arctic sea ice. The LLE method is based on selecting a certain percentage of the lowest elevation measurements within the along-track section surrounding every ICESat sample, and assumes that their mean (as in (Yi and Zwally, 2009)) or their polynomial fit (as in (Spreen et al., 2006) and (Kern and Spreen, 2015)) represents the local sea surface height for the given sample. A main limitation of the LLE method

is that in case of absence of leads or cracks in the vicinity of a given measurements or if the selected percentage of lowest elevations is larger than the actual number of measurements over leads/cracks, the level of sea surface is overestimated, and consequently the freeboard is underestimated.

The Arctic sea ice thickness from two available products derived from ICESat data by Jet Propulsion Laboratory (JPL) using the TP method (http://rkwok.jpl.nasa.gov/icesat) and by Goddard Space Flight Center (GSFC) using the LLE method (Yi and Zwally, 2009) were found to be different by 0.42 m (Lindsay and Schweiger, 2015). This difference can be caused by the different techniques for determining the local sea level in the freeboard retrieval algorithm, or by the different methods in estimating snow depth that is used when calculating ice thickness, i.e. by the uncertainty of the freeboard-to-thickness conversion (Kwok and Cunningham, 2008; Zygmuntovska et al., 2014).

In this paper we reproduce the two approaches used to retrieve Arctic total freeboard, i.e. using the TP and LLE methods. We analyze why these methods lead to differences in local freeboard estimates, show how they are distributed in space and over the ICESat period (2003-2008), and propose an improvement in the freeboard retrieval algorithm used in the TP method. The TP method presented originally in Kwok et al. (2007) was further developed and improved to take into account snow that is accumulated on thin ice in leads (Kwok and Cunningham, 2008) and size of leads with respect to the size of the ICESat altimeter footprint (Kwok et al., 2009). These two corrections were taken into account in the JPL product. Therefore we quantified their effect on freeboard estimates, and hence on the difference between the corresponding sea ice thickness products.

## 2 Data and methods

### 2.1 Data

In this study we use the ICESat level 2 data of Release 32 from 10 laser campaigns, corresponding to periods of ~35 days in autumn and winter named as 2a , 2b , 3a , 3b , 3d , 3e, 3g, 3h, 3i and 3j in the ICESat dataset and that we will hereafter denote with respect to the period covered as ON03, FM04, ON04, FM05, ON05, FM06, ON06, MA07, ON07, FM08. The abbreviations ON, FM and MA mean October-November, February-March and March-April respectively, followed by the year (i.e. 2003 to 2008). We also use along-track freeboard product derived from ICESat (Yi and Zwally, 2009) and available for download on the NSIDC server (http://nsidc.org/data/nsidc-0393).

In addition, we use the Arctic-wide multiyear ice fraction dataset used in Zygmuntowska et al. (2014) that was produced by reprocessing the QuikSCAT satellite scatterometer data. Zygmuntowska et al. (2014) used daily averaged gridded (22.5 km) data of radar backscatter processed by Brigham Young University (ftp://ftp.scp.byu.edu/data/qscat/SigBrw), and converted them into multiyear ice fraction following the method described in Kwok (2004). We also use the NSIDC, daily, 25-km-resolution sea ice concentration product based on Advanced Microwave Scanning Radiometer (AMSR-E) satellite measurements (Cavalieri et al., 2014) and available for download at https://nsidc.org/data/AE_SI25/versions/3.

## 2.2 ICESat data filtering and corrections

Unreliable ICESat elevation estimates were filtered out using waveform parameters of the altimeter returns provided together with ICESat data. When reproducing the TP and LLE methods, we used the same filtering criteria that are applied in Kwok et al. (2007), in order to compare the algorithms avoiding biases associated with different filtering. We discarded measurements where the receiver gain used for indicating forward scattering in the atmosphere (i_gval_rcv) was more than 30 and the standard deviation of the difference between received ICESat echo waveforms and the Gaussian fit (i_SeaIceVar) was more than 60. Saturated waveforms, which occurred over bright smooth flat surfaces with reflectivity (i_reflctUC: ratio between received and transmitted energy) > 1 were removed. In addition, we filtered out highly saturated returns with amplitude greater than the saturation index threshold for more than five consecutive waveform gates (i_satNdx > 5). The influence of the filtering criteria on freeboard estimates will be illustrated below in an example where we apply a different threshold for the receiver gain parameter (80, as used by Yi and Zwally, 2009).

Then we determine the elevation, $h$, above the EGM08 geoid (Pavlis et al., 2012) provided with the ICESat data, and apply saturation correction to the measurements with moderately saturated waveforms using the corresponding parameter (i_satElevCorr) and flag (i_satCorrFlg), as well as the inverse barometer correction to allow for atmospheric pressure loading (Kwok et al., 2006, Zwally et al., 2008). We discard the areas with open ocean, which we define here as the region covered by less than 30% sea ice coverage, according to the AMSR-E ice concentration product from NSIDC.

It should be noted that Kwok et al. (2007) and Yi and Zwally (2009) used ICESat data from the earlier Release 28 and estimated elevations, $h$, above the ArcGP geoid (http://earth-info.nga.mil/GandG/wgs84/agp/hist_agp. html) rather than the EGM08 geoid. Furthermore, when applying the LLE method Yi and Zwally (2009) first calculated the improved geoid before using it for freeboard estimation. Although freeboard retrieval from satellite altimetry is primarily based on estimation of local sea level in leads we evaluate the effect of using a different geoid on the results in section 3.2.

## 2.3 Algorithms used in the TP and LLE methods

In order to remove longer wavelength and large amplitude variability due to geoid, atmospheric loading and tidal errors, the first step for both (TP and LLE) freeboard retrieval algorithms is to determine relative elevations, $h_r$, defined as the difference between elevations $h$ and their 25-km (like in Kwok et al., (2007)) or 50-km (like in Yi and Zwally, (2009)) running means, $\bar{h}$, as $h_r = h - \bar{h}$ (Table 1). We evaluate and discuss the effect of the different scale of spatial smoothing when calculating $\bar{h}$ in section 3.2.

The principal distinction between the algorithms, as noted above, is the difference in the method used to determine the sea surface references. For the LLE method, Yi and Zwally (2009) assumed that the lowest 1% of the measurements along the satellite track represent elevations over open leads and therefore can be used for estimation of the local sea level. Hence, they determine elevations of sea level, $h_{sl}$, as the mean of the lowest 1% of the $h_r$ values within $\pm 50$ km around each measurement point. Therefore, in this approach the number of points used for the determination of sea level depends only on

measurements' availability. The distance between ICESat samples along track is 172 m. If we assume that all samples are reliable, 6 of about 580 measurements within the 100-km range are used to calculate the local sea surface level. However, in case there is no open water within the 100-km range, the calculated $h_{sl}$ will be the height of thin ice rather than the sea level height, leading to an underestimation of the freeboard.

For the TP method, Kwok et al. (2007) determined sea level from ICESat samples, tie-points, identified according to specified requirements. From the analysis of SAR images from RADARSAT satellite, Kwok et al. (2007) found a linear relationship between along-track elevation variability and freeboard values adjacent to new openings at the same locations where leads were identified and collocated with ICESat data. Although determination of this relationship provides a tool for detection of tie-points, the procedure of visual inspection of satellite images is time-consuming, and can be applied only for
regional analysis (e.g. Markus et al., 2011). Therefore Kwok et al. (2007) proposed to use the relationship between elevation variability and negative $h_r$ with dips in reflectivity, which are found to be associated with young ice in leads. Elevation variability is defined as the standard deviation $\sigma_{25}$ of the detrended $h_r$ within a 25-km running window. The dips in reflectivity are defined as when the difference between the local reflectivity for a given sample and the background reflectivity, $\Delta R$, is larger than 0.3. The background reflectivity is estimated as the average reflectivity of the measurements
within 25 km around a given sample that are greater than $\bar{R} - 1.5\sigma$ with $\bar{R}$ and $\sigma$ being the mean and standard deviation of reflectivity of all the measurements within 25 km around that given sample. Then they select all samples corresponding to the points below the line obtained from a regression model, here a cubic polynomial, to be used as tie-points. That is, they select samples for which $\sigma_{25}$ is less than that determined from the regression line for the given $h_r$. Kwok et al. (2007) identified two sets of tie-points: one set consists of samples located below the regression line which also have $\Delta R > 0.3$, and
the other one includes all the samples corresponding to the points located below the regression line without constraining the value of $\Delta R$. They found good agreement between two sets of tie-points as well as agreement of these sets with high-quality tie-points determined from collocation with satellite images. Since the sampling density of tie-points in the former set is not sufficient for basin-wide studies, both sets of detected tie-points are used for the calculation of sea level. The sea surface references are estimated for 25-km non-overlapping segments as an average of $h_r$ values corresponding to the tie-points,
weighted as the exponential function of the distance to the line obtained from the regression line. The higher $\sigma_{25}$, which is characteristic of the surface roughness of a given sample, the lower is the $h_r$ required to qualify this sample as a tie-point. This weighting method of the tie-points utilizes their likelihood to be a reference point, and is particularly important when many tie-points are detected within a 25-km segment. Since the position of the regression line varies over seasons and years, Kwok et al., (2007) proposed to apply the same regression model to each ICESat observation period. We discuss the
influence of the regression model in section 3.3.

**2.4 Correction of geoid**

In Zwally et al. (2008) and Kwok et al. (2007) a difference between running mean elevations, $\bar{h}$, and the determined elevation of sea level, $h_{sl}$, was found in order to characterize the unresolved residuals in the sea surface height. Since the spatial pattern of the differences is found to be consistent for different ICESat campaigns these residuals were mainly associated with the characteristics of uncertainties in the static geoid, and to a less degree as coming from time-varying components or noise in the freeboard estimation process. Therefore Yi and Zwally (2009) applied this difference, $\bar{h} - h_{sl}$, to correct geoid heights, and used this new improved geoid for retrieving the freeboard. We determined the differences $\bar{h} - h_{sl}$ for each along track measurement, and examined the effect of this geoid adjustment on freeboard estimates in section 3.2.

**2.5 Adjustments for snow depth and area of sea surface references**

After being presented in Kwok et al. (2007) the TP method for freeboard retrieval was further developed by implementing two corrections of sea surface references based on functions determined empirically. One correction is an adjustment of the elevations at the location of tie-points for taking into account the depth of snow accumulated over young ice (Kwok and Cunningham, 2008). This adjustment is based on the contrast difference in reflectivity existing between sea ice and snow surfaces, and is estimated as a function of $(1 - \Delta R)$. The correction of the sea surface references varies within the range 0 to 5 cm (Fig 2b in Kwok and Cunningham (2008)). It can be noted that increase of reflectivity over young ice may also reflect the effect of frost flowers growth. Therefore, the function of reflectivity that accounts for accumulated snow and used to determine the correction may be different in the presence of frost flowers.

Another correction accounts for the fact that ICESat measurements over tie-points are contaminated by the neighbouring sea ice surface within the laser altimeter footprint. In Kwok et al. (2009), it was proposed to multiply all the freeboard measurements by a factor of $1.1 + 0.1 \left( \frac{R_{snow} - R}{R_{ice}} \right)$, where $R_{snow} = 0.7$ and $R_{ice} = 0.25$ are the typical reflectivity of snow and ice respectively, and $R$ is the reflectivity of the ICESat measurements. This correction increases with freeboard height and decreases with the reflectivity in ICESat samples.

**3 Results and discussion**

In this section we compare different freeboard estimates retrieved using our implementations of the TP and LLE methods as illustrated on the flow chart in Figure 1. First, we test the agreement between freeboards obtained using our implementation of the LLE method and those provided in the GSFC product. Then, we analyze how the choice of different along-track averaging scales and geoid definition affect the freeboard estimates when using the TP and LLE methods, and therefore how it can partly explain the differences found between the JPL and GSFC products. We also quantify the effect of applying different approaches for determination of sea surface references in the TP and LLE methods when choosing the settings, which give consistent freeboard retrievals. Based on these analyses we propose an improvement of the freeboard retrieval algorithm used in the TP method. We also estimate the effect on freeboard estimates of applying corrections accounting for

snow depth in tie-point areas and for the size of leads, as it was done for the JPL product. Finally, we proceed with a comparison of the obtained freeboard when using the different methods and parameters, and we summarize our findings.

### 3.1 Comparison of GSFC product with freeboard retrieved using the LLE method

We checked consistency between the freeboards retrieved in this study and those available from the GSFC product by following the LLE method described in Yi and Zwally (2009): the elevations were used relative to the ArcGP geoid corrected for $\bar{h} - h_{sl}$ residuals, the calculation of $\bar{h}$ values was made using an along-track smoothing window of 50-km, and the 1% lowest elevation measurements over the 100-km along-track segment centered on each sample were used for estimation of the reference sea level $h_{sl}$. We compared our results with the freeboards of the GSFC product by first computing the differences between the freeboards calculated along track for each sample before computing their averages over a regular 25-km grid covering the data domain. Maps of freeboard estimates as well as maps of differences and their distribution for the ON05 and FM06 ICESat periods are presented in Figure 2. The mean and standard deviation of the differences for the other periods are recapped in Table 2 (first line). The mean differences are small, i.e. around ±2 cm, indicating good agreement between the estimates. The remaining discrepancies can be attributed to different data filtering, and possibly to the differences existing between data releases (e.g. improvements in saturation correction).

In particular, wide-spread underestimation of the freeboard thinner than 15 cm by up to 10 cm for the ON05 and FM06 periods (Figure 2b and 2d) can be explained by different threshold values used for the receiver gain parameter. Indeed, following Kwok et al. (2007) we used ICESat measurements with gain values of less than 30, while Yi and Zwally (2009) chose to set this threshold value to 80, thereby involving more data in their analysis. This additional portion of the data is more affected by atmospheric forward scattering, leading to measurements showing a larger range and shifted towards lower elevation values. For thin ice the likelihood is high for these elevations to be lower than the neighbouring ones along the track, and as a consequence for them to be used for determination of sea surface reference, which may finally result in higher freeboard estimates. We checked that applying the exact same threshold as in Yi and Zwally (2009) for the receiver gain parameter increases the agreement between the estimates, as one can see from the removal of the very negative (in blue) differences present over the Kara, Laptev and Chukchi Seas in the maps of Figure 2c as compared to the maps of Figure 2b.

One should note however that Yi and Zwally (2009) also used a pulse-broadening parameter for the data filtering that is not applied here. This parameter primarily depends on the width of the echo waveform and, among other effects, accounts also in part for atmospheric forward scattering. This explains why we obtain a noticeably higher freeboard as compared to the GSFC product for some ICESat periods like e.g. ON03 and ON04 when setting the threshold value for receiver gain to 80 (Table 2, second line). Thus we think that setting the threshold value for receiver gain to 30 is roughly equivalent to the filtering settings applied by Yi and Zwally (2009) to account for forward scattering, but has the advantage of being more efficient over regions of thin ice. It should be noted that Yi et al. (2011) used different thresholds for gain in order to account for the reduction of gain with the age of the ICESat's lasers due to decrease of the transmitted power. However, in this study, we compare freeboard estimates with the GSFC product, which is derived by Yi and Zwally (2009) using constant setting for

the gain threshold. The largest biases and their variability observed in the periods ON03 and ON04 correspond to the first operation periods of laser 2 (campaign 2a) and laser 3 (campaign 3a) respectively. Therefore, one may presume that the pulse-broadening parameter applied by Yi and Zwally (2009) is also affected by the instability in the power transmitted by the ICESat's lasers.

## 3.2 Sensitivity of freeboard estimates to LLE method parameters and geoid definition

The JPL and GSFC products, as noted above, are generated using different along-track averaging scales to calculate $\bar{h}$ values, i.e. 25 and 50-km respectively. The length of the along-track segment used for estimation of the local sea surface references $h_{sl}$ is also different: 1% of lowest elevations available over 100-km around each sample for the GSFC product (YI and Zwally, 2009), while 25-km non-overlapping segments are used for the JPL product (Kwok et al., 2007). Therefore, before comparing the freeboard retrievals obtained with the two methods we checked, as an example, how the choice of different averaging scales influences the results of the LLE method. Although both freeboard retrieval algorithms are based on the difference between ICESat elevations over sea ice and the neighbouring leads, which makes them almost fully independent from the geoid accuracy, we also looked at how the choice of geoid influences the results.

We compared the freeboards retrieved using the LLE method with a window's size of 25-km and 50-km to compute $\bar{h}$, and using along-track averaging segments of 25-km and 100-km to estimate $h_{sl}$. Note that these averaging scales correspond to those used to produce the JPL and GSFC freeboard estimates. As described in section 2.3 the running mean $\bar{h}$ is estimated to remove the large-scale fluctuations in the elevations caused by the geoid used. Note that in our analysis of the freeboard retrieval methods we use data from the newer EGM geoid provided with ICESat data in contrast to the ArcGP geoid used by the JPL and GSFC. Although overall effect of geoid selection on freeboard values is small, we observe some improvements after switching to the EGM geoid, which are discussed in this section below. Freeboards obtained with the longer scale (GSFC) setting exceed those obtained with the shorter scale (JPL) setting by, on average, 4 cm when applying the LLE method. (Table 2, line 3 and Figure 3a for the FM06 period). By applying other combinations of averaging scales we found that these differences depend mainly on $h_{sl}$ value. This is expected from the fact that considering a larger window increases the chance to include lower $h_r$ dips in the calculation of $h_{sl}$, although we use the same fraction of the lowest elevations (1%) for selection of tie-points. Freeboard differences due to different averaging scales for calculation of $\bar{h}$ are small and their patterns have the features of those related to geoid uncertainty shown in Figures 3b and 3c. A positive bias in freeboard estimates when using longer segments for along-track averaging, as well as tendency for enhanced biases along the coast (see Figure 3), are also reported in Kern and Spreen (2015) for the Weddell Sea in Antarctica. Freeboard differences and their variability associated with along-track averaging scales may also be linked with the surface roughness, which increases uncertainty in determination of the sea surface references. This is confirmed by looking at the period FM05 when the largest values of $\sigma_{25}$ are observed (Table 2, line 3 and Figure 6a).

The residuals $\bar{h} - h_{sl}$ applied for the correction of the ArcGP geoid are calculated using the same settings as in Yi and Zwally (2009) and shown in Figure 3d (left) for the period FM06. The spatial distribution of the residual is similar for the other periods and is in agreement with those obtained by Kwok et al. (2007). The effect on freeboard estimates is small, ranging within ±2 cm, over most of the Arctic basin (Figure 3b). The only noticeable effect on freeboard is found in the areas

of the Gakkel and Lomonosov ridges, where freeboard is reduced by about 5 cm. The areas of positive differences along the East Greenland and Canadian Arctic coasts correspond to regions of largest freeboard (Figure 2a), which is itself correlated with local surface roughness (Figure 5a), more than to the distribution of the geoid correction. The $\bar{h} - h_{sl}$ adjustment of EGM08 geoid is proportionally lower everywhere in the Arctic by about 13 cm, which corresponds to the higher level of the EGM08 geoid (Figure 3d, right). The effect of adjustment of the EGM08 geoid for $\bar{h} - h_{sl}$ on the freeboard is of the same

order in means (Table 2, line 4) and even less evident along the ridges in the central Arctic, which likely results from the overall better quality of this more recent geoid, and in particular from the better representation of small scale features. Small freeboard differences are also obtained when using the EGM08 instead of the ArcGP geoid (Table 2, line 5). In addition to the local effect along the ridges in the central Arctic the improvements in the EGM08 geoid are revealed in other areas such as along the high slopes of the bathymetric relief (see Figure 3c for the FM06 period).

Thus in order to assess the effect of different algorithms applied for determination of $h_{sl}$ in the LLE and the TP methods the same scales for along-track averaging should be used to avoid corresponding bias. In order to avoid this bias when comparing the LLE and TP methods, we chose to use an averaging window of 25-km to calculate $\bar{h}$ and $h_{sl}$ (as in Kwok et al., 2007) for the three following reasons. First, applying of the TP method using the same scales as in (Kwok et al., 2007) allow us to analyse the performance of the algorithm applied to generate the JPL product. Second, using a smaller window is

found to result in reduced dependency of the freeboard on the geoid used in the retrieval process. Indeed, in this case the correction of the geoid for $\bar{h} - h_{sl}$ as well as the fact of using a recent geoid like EGM08 no longer has any impact on freeboard along the ridges in the Arctic, as opposed to what we reported above when using larger spatial averaging. Third reason is that, as demonstrated by Kern and Spreen (2015), a more valid freeboard can be retrieved using the LLE method if the length of along-track segments considered for the selection of the lowest elevations when estimating the local sea surface

reference $h_{sl}$ is equal to, or less than, the size of the smoothing window used to calculate $\bar{h}$ values. It means that when using 100-km window for determination of $h_{sl}$ the same (or larger) scale would be preferred to calculate $\bar{h}$, so that fluctuations of the geoid would not be taken into account properly. From the other side, and as we have shown above, using a shorter length for the averaging windows when applying the LLE method results in lower freeboards that can be interpreted as underestimates due to poorer sampling of the sea surface references. The TP method, in contrast to the LLE method, is

based on selection of tie-points using a physical relationship, and its performance can be assessed by comparison of the results derived from the TP and LLE approaches. As the number of ICESat measurements available within each 25-km section does not exceed 147 samples, and most often even less due to data filtering, for each section the 1% of the lowest $h_r$ values used for estimation of $h_{sl}$ actually refer to only one sample. Kern and Spreen (2015) suggested that using such a low

percentage and consequently such a low number of samples to estimate the local sea surface reference may result in freeboard overestimation if sharp elevation changes are present along the track. However, this only happens if the size of the averaging window used to calculate $\bar{h}$ is smaller than the length of the segment used to estimate the local sea surface reference $h_{sl}$, which is not the case here since we use the same 25-km averaging scales for estimation of $\bar{h}$ and $h_{sl}$.

## 3.3 Comparison of freeboard obtained by using TP and LLE methods

### 3.3.1 The original algorithm used in the TP method

A comparison of freeboards calculated using the LLE and TP methods when applying the same along-track averaging scale as described in the previous section is presented in Figure 4. The obtained freeboard differences are small on average, ranging within ±5 cm, while the presence of significant regional discrepancies should be noted. Since the maps in Figure 4 show a clear distinction between the differences over thin and thick ice for some of the ICESat periods, we estimated differences between freeboards separately for the first-year ice (FYI) and multi-year ice (MYI) regions over the same 25-km grid cells. Grid cells are considered as covered with FYI or MYI according to the 50% isopleth on the multi-year ice fraction maps that were derived by Zygmuntowska et al. (2014) from QuikSCAT scatterometer following the method described in Kwok (2004).

Negative differences, which correspond to lower freeboard being retrieved using the LLE method as compared to the TP method according to the convention used here, are found for areas covered by MYI and located north of Greenland and Canadian Arctic Archipelago. The largest negative differences are observed for the period FM08 and are about -15 cm. These can be explained by the fact that in these areas of thick compact ice the TP method does not detect any leads for many 25-km segments, while the LLE method provides freeboard estimates because it calculates a local sea surface reference from the 1% lowest elevations that are not necessarily representative of local sea level. One can also expect difference between the basin-wide freeboard means in the area of thick MYI to be even larger when the TP method does not detect any tie-points within some grid cells and, hence, does not provide freeboard estimates. We therefore conclude that the difference found between the gridded mean freeboard retrieved by these two methods in areas of MYI is coming from (i) the lower-biased estimates due to using the measurements over refrozen leads or ice within the 25-km range for the calculation of local sea surface references in the LLE method or (ii) the absence of local detection of tie-points in the TP method, mainly where the ice cover is continuous, i.e. with presence of only few or no leads.

Positive differences are obtained over large areas of FYI and thin part of MYI for most of the ICESat campaigns, with a peak in FM08. The mean differences over FYI are within 3-5 cm, while locally these can be more than 10 cm. Since the lead fraction in the areas of seasonal ice is higher than over thick MYI (Willmes and Heinemann, 2016; Ivanova et al., 2016; Röhrs et al., 2012; Brohan and Kaleschke, 2014), one can expect that the observed difference in freeboard estimates is not coming from the same reasons mentioned above for MYI areas. The positive differences are most likely reflecting the underestimation of freeboard retrieved by the TP method as was found by Kwok et al. (2007) from comparison with the

freeboards adjacent to leads detected on satellite images and collocated with ICESat data. They showed that the freeboard underestimation was on average of 1.3 to 4 cm for ON05 and FM06 periods, and explain this by the fact that samples, which are identified as tie-points, do not always represent open water or the thinnest ice in leads. In order to explain why freeboard differences are observed primarily over FYI areas we investigated the performance of the algorithm used in the TP method.

The results are presented in the next section.

### 3.3.2 An improved algorithm for the TP method

In the TP method, as described in section 2.3, one has to find the samples that will be used as tie-points. To do so, we first establish the relationship between $h_r$ and $\sigma_{25}$ using a regression model for the measurements showing dips in reflectivity (a cubic polynomial function in Kwok et al., 2007). Figure 5 shows the relationship found between $h_r$ and $\sigma_{25}$ for the ten

ICESat campaigns. We note that visually the curves corresponding to ON05 and FM06 periods are in agreement with those reported in Kwok et al. (2007). According to Kwok et al. (2007), after the relationships between $h_r$ and $\sigma_{25}$ are established, the tie-points can be defined by taking the samples found to be below the regression lines. The sea surface reference for each 25-km segment is estimated by averaging the $h_r$ values corresponding to the tie-points, weighted exponentially by the distance from the regression line (Kwok et al., 2007). Hence, the contribution of tie-points with larger distances dominates

when calculating the local sea surface reference. The $\sigma_{25}$ values are smoothed along-track and do not change remarkably over the segment, while $h_r$ may vary significantly from sample to sample. The tie-points with lower $h_r$ contribute more than those with larger $h_r$ for a given $\sigma_{25}$. However, as shown by the flattening of the curves in Figure 5, the quasi correlation existing between $h_r$ and $\sigma_{25}$ is lost as it can be seen from the flattening of the curves in Figure 5. Although using a cubic polynomial fit of the data as in Kwok et al. (2007) reduces this flattening, the correlation does not hold towards zero $h_r$ for

many ICESat periods. As seen from the averaged regression lines on the Figure 5 a deviation from the linear relationship is more pronounced for the winter periods and starts in a freeboard range from –15 cm to –20 cm. However, we think that this flattening of the curves may not represent an actual and physically-based relationship existing between $h_r$ and $\sigma_{25}$. If this is the case, some samples may be unreasonably identified as tie-points in the TP method due to enlarged area below the regression line at $h_r$ close to zero. The effect of the flattening of the curves on the result of the regression model is illustrated

in Figure 7 for two winter periods: FM05, when the highest $\sigma_{25}$ values are seen for $h_r$ close to zero, and FM08, for which the largest discrepancy between LLE and TP results is observed. The computed regression line for the FM05 period reduces the inversed correlation between $h_r$ and $\sigma_{25}$, but still deviates from the linear relationship (see red dashed lines in Figure 7a). In FM08 the curve is less noisy but correlation is even inversed with an inflection point around $h_r = -5$ cm. As a consequence, the samples detected as tie-points that have a value of $h_r$ close to zero (i.e. measurements taken over areas covered by

thinner ice) may contribute more than the other, leading to an artificial increase of the reference sea level height over given segments. The fact that the linear relationship between $\sigma_{25}$ and $h_r$ does not hold for small freeboard can be explained by a lower likelihood for samples with $\Delta R > 0.3$ to represent actual leads. This is illustrated by the increase of the standard

deviation of $\sigma_{25}$ and the decrease in number of samples used in evaluating $\sigma_{25}$ for low absolute values of $h_r$ (Figure 7b and 7c, red). Note that this is consistent with the more pronounced flattening obtained for the winter periods, when variability of the surface roughness is larger.

Underestimation of the freeboards retrieved over thin FYI by the TP method as compared to those retrieved from the LLE

method increases with the number of samples detected as tie-points and with surface roughness. This can be explained from the fact that a large number of tie-points or a high degree of roughness increase the chance that some of those tie-points would be associated with the flattening part of the curve relating $\sigma_{25}$ and $h_r$. In general, the number of tie-points and roughness are anti-correlated and their spatial patterns match very well with the pattern of multi-year versus first-year ice as shown in Figure 6 for some selected ICESat periods discussed in the text, i.e. FM05, ON05, FM06 and FM08. The number

of detected tie-points within each 25-km non-overlapping segments ranges from a few tie-points (i.e. < 10) over MYI to several tens (i.e. > 25) over FYI (Figure 6b) and shows a significant spatial variation. A surface roughness represented by $\sigma_{25}$ (Figure 6a) typically does not exceed 10-15 cm over FYI, although it may be locally more than 20 cm as for FM05 period, when the largest roughness is observed (Figure 6a). Therefore, the differences of freeboard estimates in FM05 are primarily related to the surface roughness, and less to the number of detected tie-points, which is comparatively low. In

contrast, the large number of detected tie points plays a key role in FM08, while surface roughness over FYI is low.

In order to reduce such bias and to ensure that the selected samples used to establish the relationship between $\sigma_{25}$ and $h_r$ are actually over leads we propose an improvement to the TP method. This improvement is based on further constraining the method of selection of samples by requesting that dips in both reflectivity and elevation need to be actually measured. Here, we select samples with $\Delta R > 0.3$ and $h_r < \overline{h_{r25}} - 0.5\sigma_{25}$, i.e. samples where $h_r$ deviates from the 25-km running mean $\overline{h_{r25}}$

by at least half of a standard deviation. For $h_r < -15$ cm the resulting relationships between $\sigma_{25}$ and $h_r$ is very similar to the one obtained for the previous selection of samples, while for $h_r > -15$ cm both the high variability and inverse distribution are removed (Figure 7a, black). Since applying additional requirements on the selection of samples considered in the data regression reduces their number, especially for near zero $h_r$, we only consider as reliable the (1-cm) $h_r$ bins for which $\sigma_{25}$ is estimated from at least 15 samples. It should be noted that despite actually lower number of samples selected with this new

method the variability of $\sigma_{25}$ is significantly decreased, and the relationship between $h_r$ and $\sigma_{25}$ over thin ice remains robust over the whole range of $h_r$ and $\sigma_{25}$ values. Note that we also tried to apply more stringent selection requirement on the elevation dips like e.g. $h_r < \overline{h_{r25}} - \sigma_{25}$ and $h_r < \overline{h_{r25}} - 1.5\sigma_{25}$. In this case, the resulting $\sigma_{25} = f(h_r)$ relationships (Figure 7a cyan and blue lines, respectively) are shifted downward compared to the previous one obtained when requiring $h_r < \overline{h_{r25}} - 0.5\sigma_{25}$ (Figure 7a, black line). These lines represent rather the relationships of the mean $\sigma_{25}$ that Kwok et al. (2007)

obtained using collocation of the satellite images with ICESat data, which we mentioned above in the section 2.3. From this analysis, using the condition $h_r < \overline{h_{r25}} - 0.5\sigma_{25}$ in this improved TP method appears to be the most appropriate because it corrects the relationships for thin ice and, at the same time, better reproduces the TP algorithm for $h_r < -15$.

We tested our new TP method on the whole ICESat dataset using the additional constrain $h_r < \overline{h_{r25}} - 0.5\sigma_{25}$ in the procedure of selection of samples used to form relationships between $h_r$ and $\sigma_{25}$. The difference between the freeboards retrieved with the new TP and LLE methods (Figure 8) is now largely reduced over FYI. Depending on the period considered, the mean difference is now varying from 1.5 cm to 3.1 cm (1.6 cm to 3.3 cm for FYI) as compared to 2.4 cm to

4.4 cm (2.5 cm to 5.3 cm for FYI) before, while the range of standard deviation of the differences remains similar. The most remarkable improvement is observed for the FM08 period when the difference of 5-10 cm over vast areas of FYI are reduced to differences ranging within ±2 cm. As expected, the differences remain almost unchanged for MYI since our modification of the TP method primarily impacts freeboard estimate over thin ice areas.

### 3.4 Impact of snow depth in leads and lead size adjustments on sea surface reference calculation

Corrections to account for snow depth at the location of tie-points (which are supposedly leads) and for the size of leads with respect to the size of ICESat footprint (as proposed and applied by Kwok and Cunningham (2008) and Kwok et al. (2009)) is another source of contribution to the differences between the sea ice thickness products from the JPL and GSFC (see section 2.5). The adjustment of freeboard included in the TP algorithm and related to snow depth in refrozen leads is limited to 5 cm and is about +2-3 cm on average for all the periods considered (Table 3, first line). Due to the fixed limit, this correction is

rather uniformly distributed over the Arctic although we note that the lowest values are observed for thin ice in the Arctic seas, i.e. in the warmer regions with slower initial ice growth in leads (Figure 9a for ICESat periods ON05 and FM06). The other adjustment of freeboard related to the fact that lead area does not cover the entire ICESat footprint at the locations where tie-points are detected is applied after the adjustment for snow depth previously mentioned. The magnitude of that second correction is primarily correlated to freeboard height and ranges from +3 to +7 cm on average over the Arctic

depending on the period considered. Indeed, we observe that this correction is less important for the ICESat periods from and after ON05, i.e. when sea ice thickness starts to reduce significantly (Table 3, second line). Example of spatial distribution of that correction for the periods ON05 and FM06 is shown in Figure 9b. We can see that the largest corrections are observed over MYI areas. Depending on the ICESat period considered, the mean of that correction varies from 3.7 to 9.3 cm and from 2.3 to 5.6 cm over MYI and FYI areas, respectively.

The sum of these two corrections is about +7 cm on average over the Arctic and over the ICESat period, and is ranging from 5 to 10 cm depending on the particular period. Note that the mean corrections reported in Table 3 are estimated using the original TP method, and that we found very similar results when using the improved TP method that we propose in section 3.3.

In principle, the freeboards derived by the LLE method can also be corrected for snow depth in leads and lead width, but this

was not done for the GSFC product. Although the LLE method selects only the lowest elevations to determine the local sea level, these samples may be contaminated by snow accumulated in leads or by the neighbouring sea ice surface within the laser altimeter footprint. As shown above from the comparison of freeboards derived by the LLE and TP methods, the LLE method has a weakness mostly over the thickest ice due to lack of leads. The GSFC product, in addition, was derived using

longer averaging windows that, in general, increases a likelihood for the lowest elevations to represent a true height of sea level. Therefore, we think that when using the LLE method the empirical functions proposed by Kwok and Cunningham (2008) and Kwok et al. (2009) should be modified depending on the averaging scale applied. Otherwise, application of these corrections would yield an increase in freeboard of the same order as for the freeboards retrieved by the TP method (not shown).

## 3.5 Summary

Mean freeboard calculated over the whole arctic basin using different methods is shown in the Figure 10. The figure includes freeboard estimates from the GSFC product (black), estimates we calculated using the same LLE method but with finer resolution along-track averaging (red), those we calculated using the original TP method as described in Kwok et al. (2007) (green), and those calculated using the improved TP method we propose in this study, with (blue) or without (green dashed) adjustments for snow depth and leads width. Note that these results correspond to the area considered in the JPL product, i.e. to the Arctic Ocean without surrounding Arctic seas such as Greenland Sea, Barents Sea, Kara Sea and Baffin Bay. Correspondingly, the difference of 0.42 m between sea ice thickness in the JPL and GSFC products found by Lindsay and Schweiger (2015) was derived using randomly selected samples over Arctic Basin. Although excluding of the above-mentioned seas does not significantly impact the difference between the results, the freeboard means are changed within the range of ±2 cm depending of the ICESat period. The average freeboards obtained from the different methods and corresponding to the results shown on Figure 10 are recapped in Table 4.

Because of the use of different averaging scales to calculate sea surface references, the freeboards we estimated using the LLE method with a 25-km averaging window are lower by ~3 cm on average as compared to those of the GSFC product for all ICESat periods (Figure 10, red and black lines). As we discussed in section 3.2, this can be explained by the increased likelihood for the lowest elevations to correspond to actual sea level height when using a larger window (e.g. 100 km as applied to produce the GSFC dataset.

The freeboards we estimated using the original TP method are lower by ~3 cm on average as compared to those we obtained with the LLE method with identical 25-km along-track averaging scales (Figure 10, green and red lines). As already shown by Kwok et al. (2007), this study shows that the tie-points for determination of sea surface references selected by the TP method not always represent open water or the thinnest ice in leads. Although the tie-points with lower elevations have greater weight in most cases, the resulting sea surface reference is biased positive, hence leading to lower freeboard estimates (see Figure 5 and 7). Therefore, we suppose that obtained difference between the TP and LLE results reflects the underestimation of the freeboard derived by the TP method. The lower freeboard obtained by the TP method is mostly observed over FYI and part of MYI in the central Arctic especially in the first three ICESat periods (ON03, FM04 and ON04) (Figures 4 and 10). However, the difference between freeboards retrieved by the LLE and TP methods is reduced by more than 30% for the whole Arctic and by 40% for FYI when applying our suggested improvements of the algorithm used in the TP method. At the same time, over thick MYI the LLE method tends to give lower freeboards. Although it is not

reflected in the mean values on the Figure 10, it can be seen on the maps of the differences (Figure 4 and 8), especially for the FM08 period. This is consistent with our expectations that over thick part of continuous MYI with fewer leads, the use of a relationship between the freeboard and surface roughness for identification of tie-points, as done in the TP method, gives more reliable freeboard estimates.

The adjustments for snow depth in leads and lead width were applied only when producing the JPL dataset, and we estimated that their combined effect increases freeboard by about 7 cm. Although the samples used for determination of sea level in the LLE method can also be affected by snow accumulation and contaminated by the neighbouring sea ice surface, these corrections were not applied to freeboards in the GSFC product. Therefore, since the ratio between mean total freeboard and thickness reported in Kwok et al. (2009) is about 6, the application of these two adjustments could be in

principle sufficient to explain the difference of 0.42 m on average found between the JPL and GSFC sea ice thickness products (Lindsay and Schweiger, 2015). However our results show that after applying both corrections the freeboards retrieved using the original and improved TP method are, on average, similar and higher by ~1 cm respectively, as compared to the GSFC product.

As sea ice freeboard data from JPL are not available we cannot check their consistency with those estimated for this study

using the same method, or provide a comparison with the estimates provided in the GSFC product. According to our findings the freeboards of the GSFC product and those that were most likely calculated at JPL are close on average, meaning that the difference between the JPL and GSFC averaged sea ice thicknesses are probably coming from the difference in the choice of parameter values used in the freeboard-to-thickness conversion.

## Conclusions

In this paper we reproduced two methods already used in other studies to retrieve total (sea ice plus snow) freeboard using ICESat data, called the lowest level elevation (LLE) and tie-points (TP) methods. The main difference between these two methods reside in the different ways to determine the local sea surface reference – a key step in the process of estimating freeboard. Two available products of the Arctic sea ice thickness, GSFC and JPL, were derived respectively from freeboards retrieved with these two approaches, but were found to differ significantly, i.e. by 0.42 m (Lindsay and Schweiger, 2015). In

this study we analyzed the possible reasons for freeboard discrepancies when using the LLE and TP methods as well as their contribution to the observed thickness difference.

We first reproduced the freeboard estimates of the GSFC product by using the algorithm of the LLE method. We estimated the contribution of using different along-track averaging scales in the TP and LLE methods (as it is the case between Kwok et al. (2007) and Yi and Zwally (2009), respectively) on the freeboard estimation and how it could possibly explain the sea

ice thickness differences found between the JPL and GSFC products. The along-track averaging scales used by Yi and Zwally (2009) are larger than those used by Kwok et al. (2007), resulting in higher freeboard estimates by ~3 cm on average. We also estimated the effect of the geoid adjustment applied by Yi and Zwally (2009) for the residuals between the geoid

heights and the sea level determined from ICESat data. We found that noticeable freeboard differences are observed only locally, i.e. along the high slopes of the bathymetric relief, and only when using large along-track averaging scale.

In order to analyse the effect of using different approaches to estimate local sea surface references, the same 25-km along-track averaging was applied for both the LLE and TP methods. We showed that locally and over thick and continuous MYI
cover areas the LLE method gives a lower freeboard by up to 15 cm when compared to the TP method. Over FYI, in contrast, the LLE method gives freeboards that are higher by 3-5 cm on average compared to the TP method. This is explained by the fact that ICESat samples selected for calculating local sea level do not always represent the lowest elevations and by their inadequate weighting in these calculations when applying the TP method. We proposed an improvement in the algorithm of the TP method that results in a much better agreement over FYI with the LLE method, i.e.
with differences reduced to less than 2 cm on average. Since it is based on a physical relationship and seems adequate to give reasonable results over both MYI and FYI areas, we therefore recommend using the TP method with our improved algorithm over the LLE method to calculate local sea surface references.

The freeboard corrections that have been applied in the JPL product to account for snow depth in leads and for lead width with respect to the size of the ICESat altimeter footprint (Kwok and Cunningham, 2008; Kwok et al., 2009) are significantly
impacting the freeboard values estimated using the TP method, accounting for an increase of about 7cm on average.

Overall, we showed that the different along-track averaging scales and approaches to calculate sea surface references, from one side, and the freeboard adjustments as applied in the TP method used to produce the JPL dataset, from the other side, are roughly compensating each other with respect to freeboard estimation. Indeed, we obtain similar freeboard estimates while using the TP and LLE methods with the set of parameters used by Kwok et al. (2009) and Yi and Zwally 2009, respectively.
We therefore suspect that the differences found in the JPL and GSFC sea ice thickness products are not intrinsically due to the difference in the freeboard retrieval methods, but may be attributed to the use of differences in the freeboard-to-thickness conversion.

In conclusion, we show that using different methods for freeboard retrieval from ICESat data leads to wide-spread differences between freeboard estimates over large areas in the Arctic. In particular, significant freeboard biases are found when using
different algorithms and averaging scales for determination of the sea surface height, to which the freeboard is referenced. These biases depend on sea ice characteristics, such as lead fraction and surface roughness, and therefore vary in space and time. In addition, the freeboard adjustments accounting for snow depth in leads and lead width in the JPL product significantly affect freeboard values. However, it is difficult to assess the validity of the empirical relationships proposed for calculation of these corrections by Kwok and Cunningham (2008) and Kwok et al. (2009) and whether they should be
applied to the freeboards retrieved by the LLE method using larger averaging scale as it was done for the GSFC product. We demonstrate and quantify the sources of uncertainties in the freeboard retrieval process, and propose the improvement of the TP method that can be used for further studies related to the freeboard retrieval from satellite laser altimetry. Although ICESat-1 is currently not in orbit anymore these findings can be used in the future studies for analysis of the data from the follow up ICESat-2 satellite planned for launch in 2018.

**Acknowledgments.** This work was funded and supported by the Nansen Environmental and Remote Sensing Center and by the ESA Climate Change Initiative, Sea Ice project (SICCI). Authors would like to thank Stefan Kern and the anonymous referee for their numerous comments that greatly helped to improve the manuscript.

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

Table 1. Methods and settings used for freeboard retrieval in the GSFC and JPL products.

| Institute | Method to estimate $h_{sl}$ | Segment length for $\bar{h}$, km | Segment length for $h_{sl}$, km |
|-----------|------------------------------|----------------------------------|----------------------------------|
| GSFC | LLE (lowest 1%) | 50 | 100 |
| JPL | TP (using $\sigma_{25} = f(h_r)$) | 25 | 25 |

Table 2. Mean±std of the differences between freeboards estimated using different averaging scales and geoids (cm). The compared methods are indicated by the used geoid (ArcGP or EGM08), geoid adjustment is indicated by dif, scale of along-track averaging is indicated by 100 (i.e. applying of 50-km and 100-km windows for estimating of $\bar{h}$ and $h_{sl}$) and 25 (i.e. applying of 25-km windows for estimating of $\bar{h}$ and $h_{sl}$).

| Methods compared | ON03 | FM04 | ON04 | FM05 | ON05 | FM06 | ON06 | MA07 | ON07 | FM08 |
|---|---|---|---|---|---|---|---|---|---|---|
| ArcGP/dif/100 – GSFC | 0.3±6.5 | 0.4±5.6 | 0.6±6.7 | 1.2±7.3 | -1.5±6.6 | 0.1±5.7 | -1.7±6.5 | -0.3±5.9 | -2.0±7.2 | -0.4±5.7 |
| ArcGP/dif/100 (gain 80) – GSFC | 4.6±12.4 | 1.3±5.5 | 4.6±9.2 | 3.1±8.6 | 1.9±6.3 | 1.7±5.5 | 1.4±5.5 | 1.5±5.5 | 2.8±6.0 | 1.2±5.2 |
| EGM08/100 – EGM08/25 | 5.2±8.8 | 3.6±7.3 | 5.7±8.4 | 6.4±9.1 | 3.7±7.8 | 3.8±7.5 | 3.3±6.6 | 3.8±7.7 | 4.1±8.4 | 3.8±8.6 |
| EGM08/dif/100 – EGM08 / 100 | 0.1±5.2 | 0.7±4.2 | 0.0±5.2 | 0.2±7.0 | 0.3±4.8 | 0.7±3.9 | 0.3±4.6 | 0.5±4.4 | 0.0±4.7 | 0.5±3.8 |
| EGM08/100 – ArcGP /100 | 0.0±3.7 | 0.2±2.7 | 0.0±3.2 | 0.2±2.7 | 0.1±3.0 | 0.2±2.8 | 0.1±3.1 | 0.2±2.7 | 0.0±5.9 | 0.2±2.7 |

Table 3. Mean±std of the adjustments of freeboard retrieved by the TP method to account for snow depth in refrozen leads and for lead width with respect to the size of the ICESat footprint (cm). The adjustments are estimated following the methods described in Kwok and Cunningham (2008) and Kwok et al., (2009).

| Correction | ON03 | FM04 | ON04 | FM05 | ON05 | FM06 | ON06 | MA07 | ON07 | FM08 |
|---|---|---|---|---|---|---|---|---|---|---|
| Snow depth | 2.5±1.0 | 3.0±1.0 | 2.8±1.0 | 3.0±1.0 | 2.7±1.1 | 2.7±1.1 | 2.7±1.2 | 2.9±1.2 | 2.3±1.3 | 2.6±1.1 |
| Lead width | 4.5±2.3 | 3.1±1.7 | 6.1±3.1 | 6.9±3.3 | 4.1±2.4 | 4.1±2.2 | 3.3±1.9 | 3.6±1.9 | 3.2±1.8 | 2.6±1.6 |

Table 4. Mean freeboard as derived from the GSFC product and estimated in this study using different methods, along-track averaging scales, and with or without applied corrections for snow depth in leads and for lead width.

| Area | GSFC | LLE, 100 km | LLE, 25 km | TP | TP + corrections | TP modified | TP modified + corrections |
|------|------|-------------|------------|-----|------------------|-------------|---------------------------|
| Overall | 32.7 | 33.0 | 29.5 | 26.4 | 33.4 | 27.3 | 34.5 |
| FYI | 24.2 | 24.4 | 20.7 | 17.6 | 23.5 | 18.9 | 24.9 |
| MYI | 43.4 | 44.3 | 40.3 | 38.5 | 46.9 | 38.8 | 47.3 |

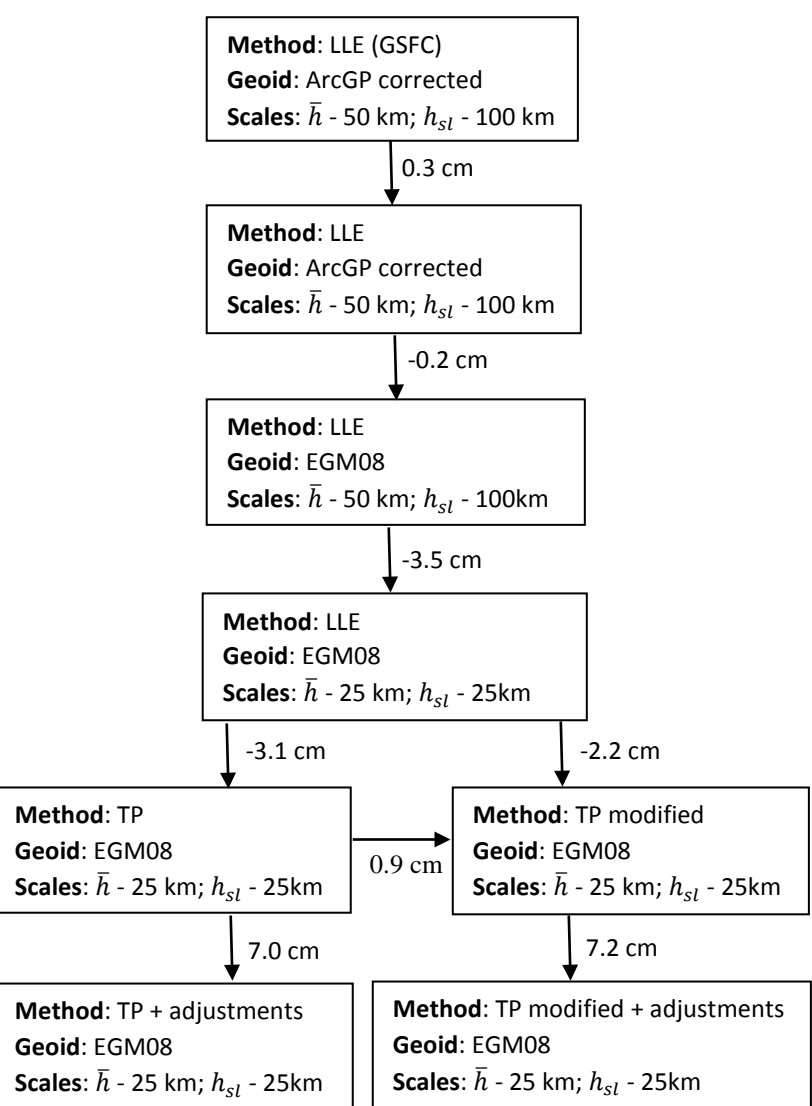

Figure 1. Flow chart of the methods, settings and geoids used for freeboard retrieval in our implementations of the TP and LLE methods. Values next to the arrows are the mean differences between freeboard estimates as derived from the Table 4.

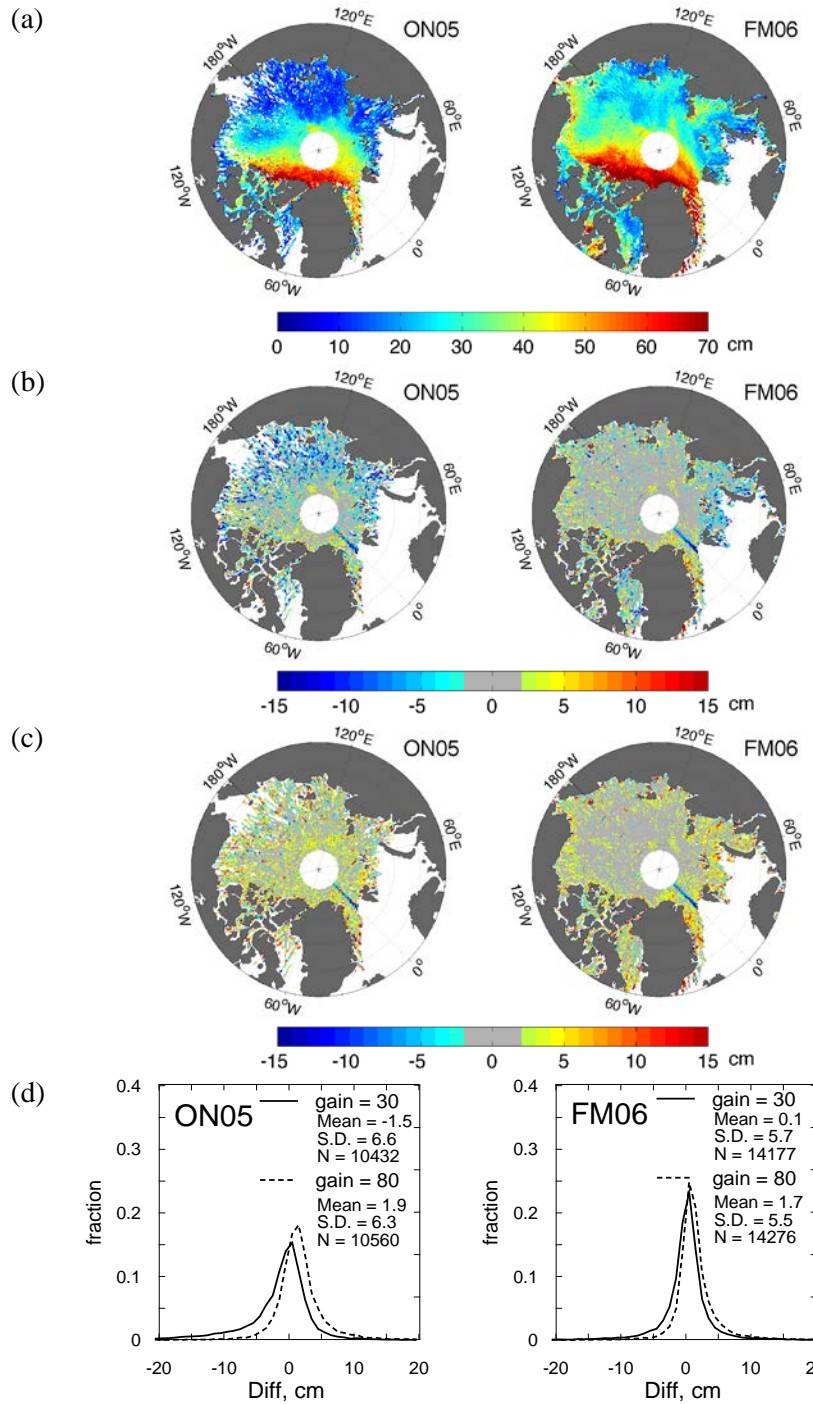

Figure 2. (a) Freeboard retrieved by the LLE method using along-track averaging scales as applied by Yi and Zwally (2009) to derive the GSFC product and (b-d) its differences from the GSFC freeboards (freeboads from this study minus GSFC freeboards) for ON05 and FM06 periods gridded into 25-km bins (cm). The freeboards estimated in this study are obtained using ICESat data with receiver gain of smaller than 30 (b) and 80 (c). (d) Distribution of the differences between freeboards. An artefact line of negative differences along the 0 longitude in (b) and (c) is due to an unexplained positive anomaly in the GSFC freeboard estimates.

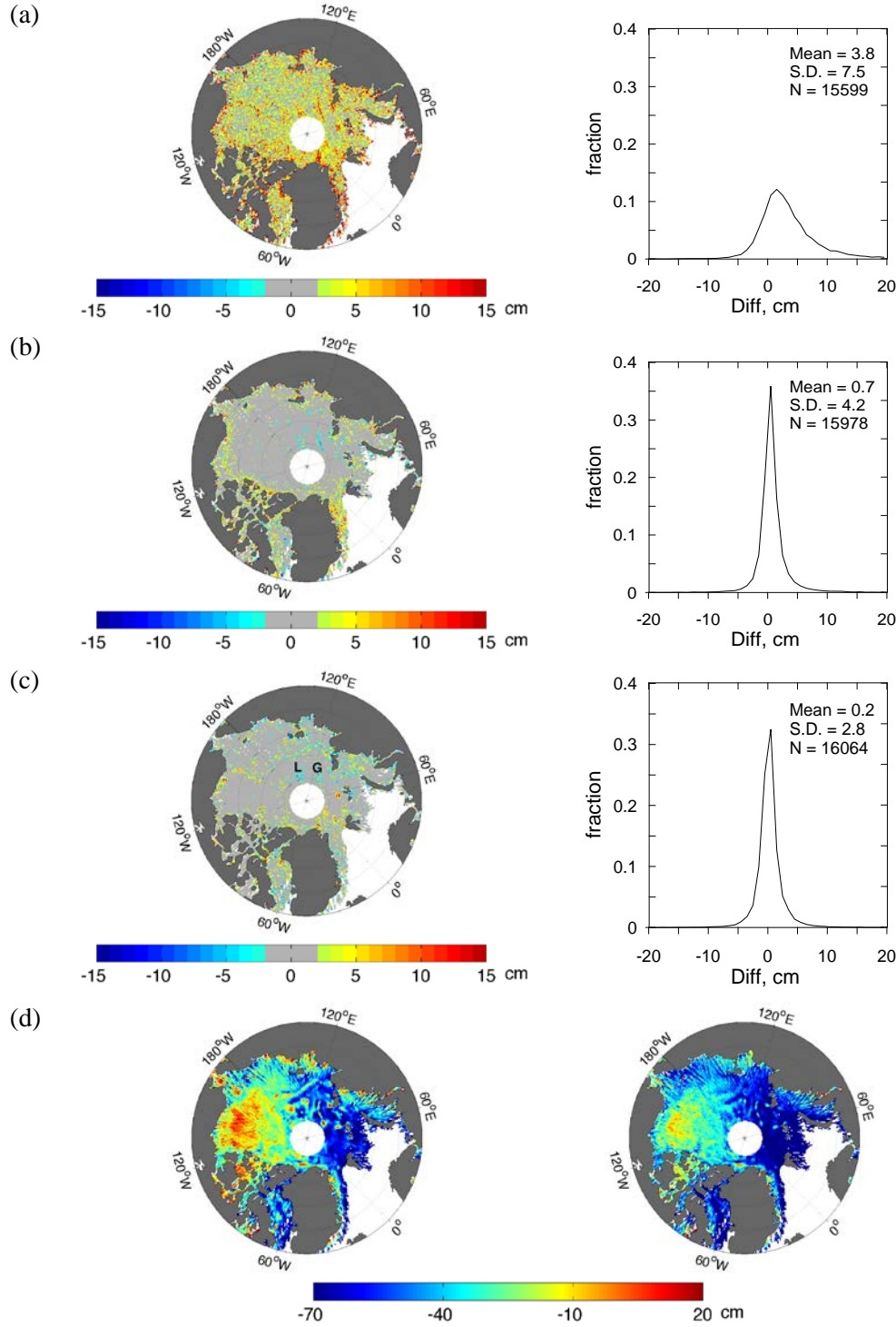

Figure 3. Effect of using different geoids and scales for along-track averaging on freeboard estimates (cm) when applying the LLE method for FM06 period (25-km grids). Differences between freeboard estimates show the effects of (a) applying longer and shorter along-track averaging scales (longer minus shorter) when using EGM08 geoid as well as (b) adjustment of ArcGP geoid for $\bar{h} - h_{sl}$ values (adjusted minus unadjusted) and (c) using different geoids (EGM08 minus ArcGP) in case of applying longer along-track averaging scales. (d) Adjustment for $\bar{h} - h_{sl}$ to correct ArcGP (left) and EGM08 (right) geoids when using coarser resolutions (cm). G and L in (c) point out the location of the differences along the Gakkel and Lomonosov ridges.

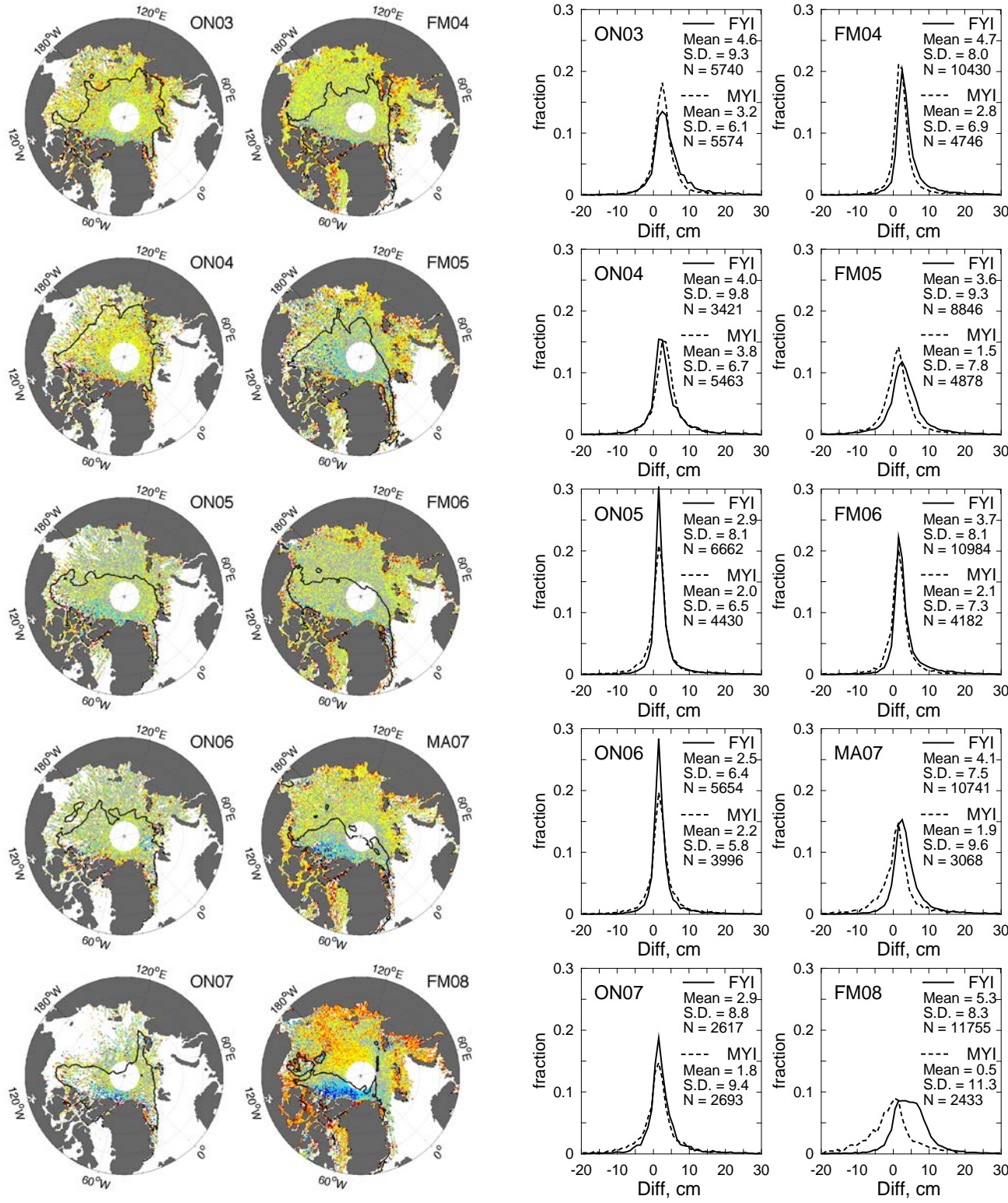

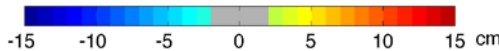

Figure 4. Maps (left) and distributions (right) of the differences between freeboard (25-km grids) estimated using LLE and TP methods (LLE minus TP) for ten ICESat periods (cm). Thick black line on the maps delineates the average 50% isopleth of multiyear ice fraction.

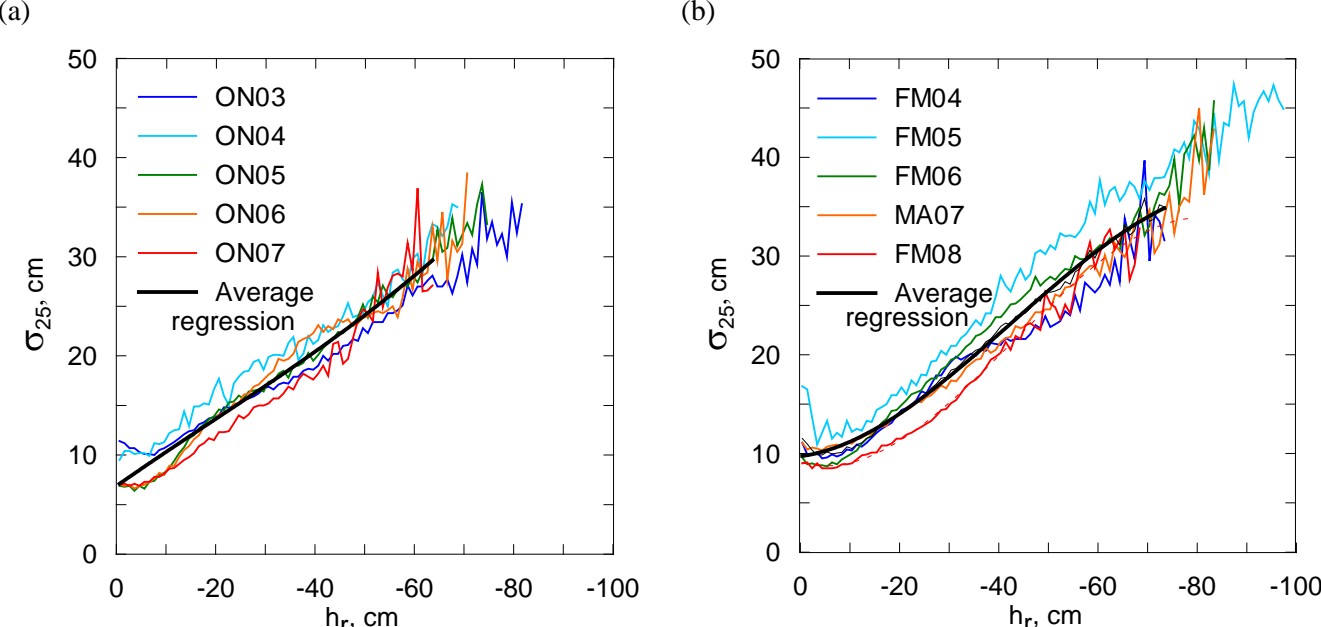

Figure 5. The relationships between $h_r$ and $\sigma_{25}$, for samples where dips in reflectivity is measured for fall (a) and winter (b) ICESat periods, following the method described in Kwok et al. (2007). The $h_r$ axis is discretized in bins of 1-cm. Note that following Zwally et al. (2008) and Yi and Zwally (2009) we define $h_r$ as $h_r = h - \bar{h}$ and form the relationship for negative $h_r$ values, while in Kwok et al. (2007) $h_r = \bar{h} - h$ and positive $h_r$ values are considered.

(a)

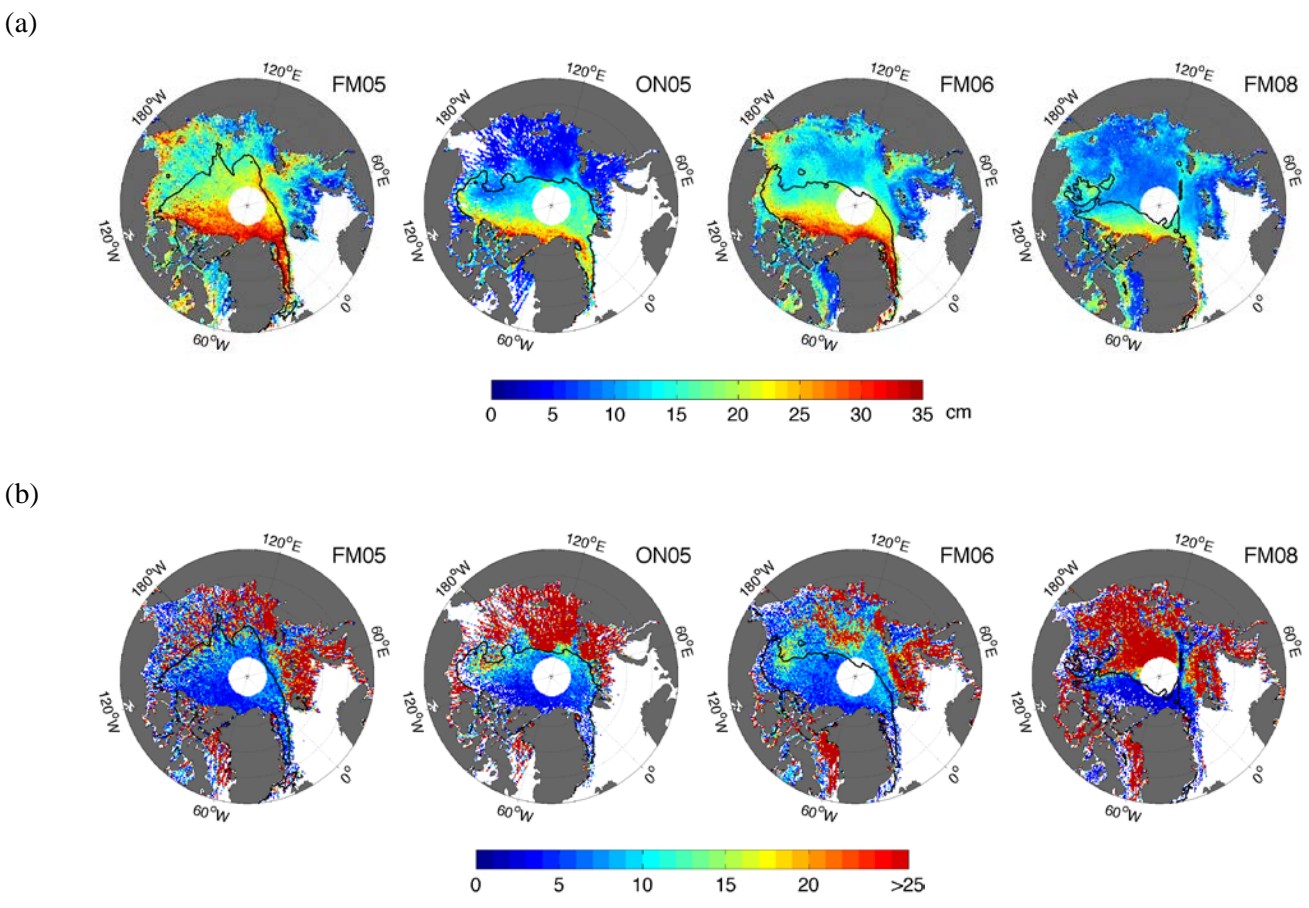

(b)

Figure 6. Standard deviation of detrended elevations $h_r$ (cm) (a) and number of tiepoints within 25-km non-overlapping segments detected by the TP method (b) for FM05, ON05, FM06 and FM08 periods.

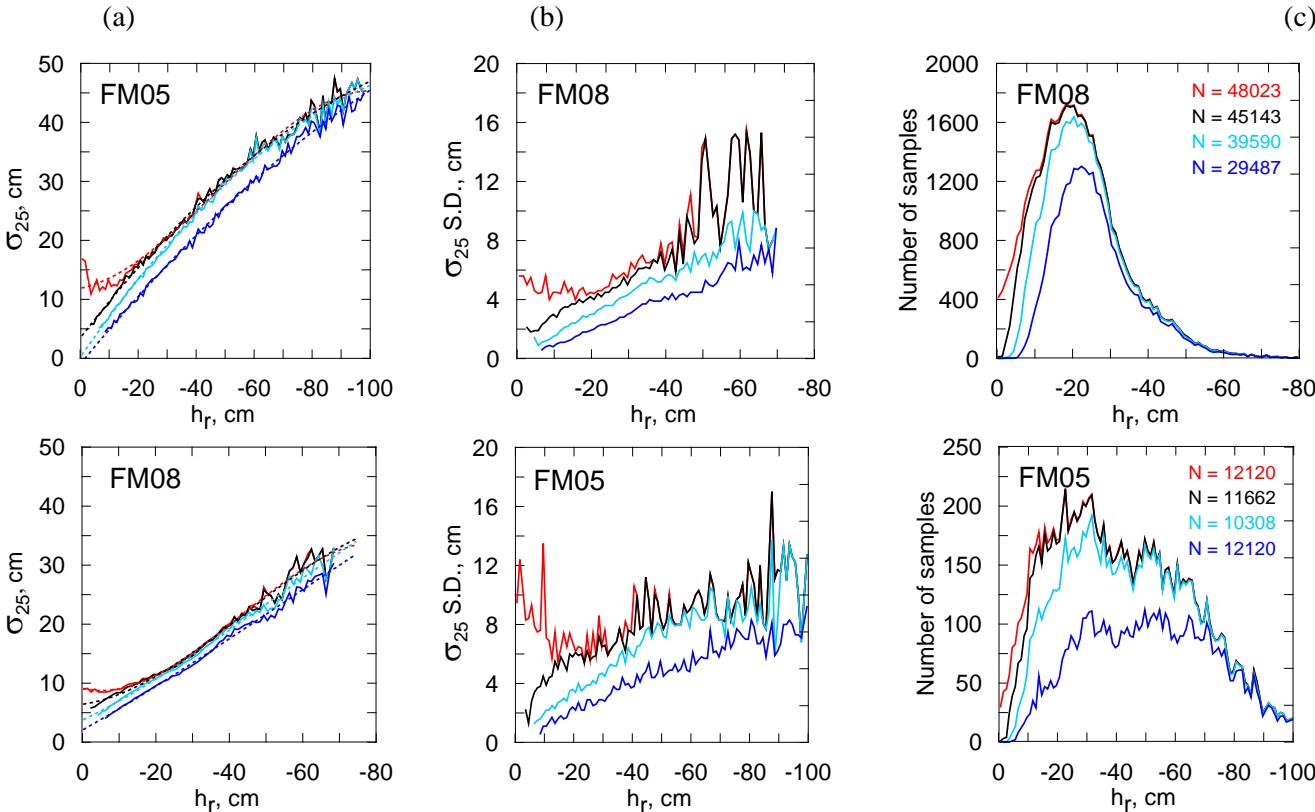

Figure 7. The relationship between $h_r$ and $\sigma_{25}$ (a), distributions of its standard deviations (b), and number of samples in each $h_r$ bin (c) for FM05 and FM08 ICESat periods. Red lines are constructed from the selection of samples for which dips in reflectivity are measured, as described in (Kwok et al., 2007). Black, cyan and blue lines are constructed from the new selection of samples we propose in this study, based on requesting the presence of dips in both reflectivity and elevation measurements. Dashed lines correspond to the result of the regression model (cubic polynomial fits) applied to the data (See text for details).

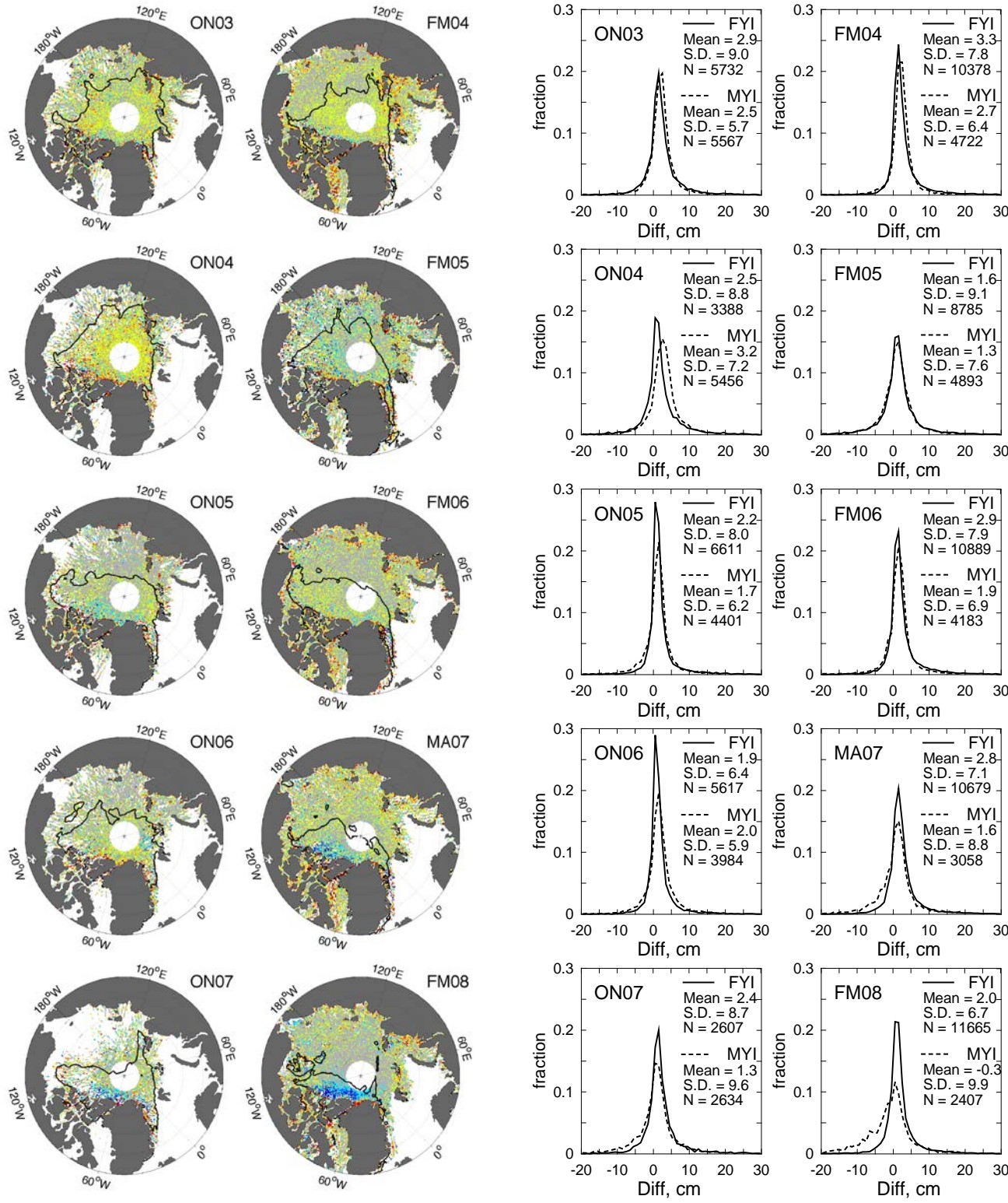

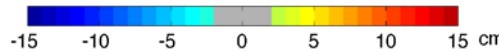

Figure 8. Maps (left) and distributions (right) of the differences between freeboard (25-km grids) estimated using LLE and TP methods (LLE minus TP) for ten ICESat periods (cm). The TP method used here includes the improvements in the freeboard retrieval algorithm proposed in this study. Thick line on the maps delineates the 50% isopleth of multi-year ice fraction.

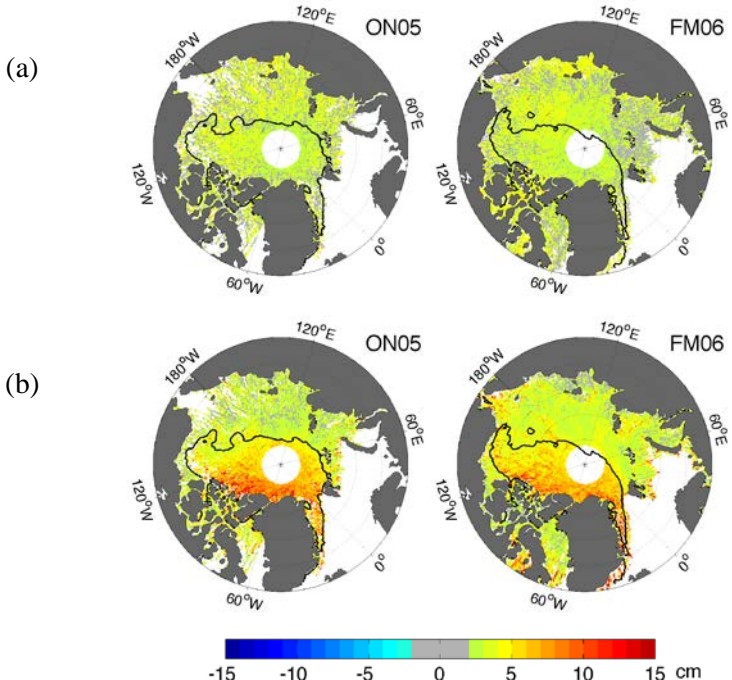

Figure 9. Adjustments of freeboard retrieved using the TP method accounting for (a) snow depth on top of new ice in leads and (b) lead width with respect to ICESat footprint area for the periods ON05 and FM06. These adjustments are estimated following the methods described in Kwok and Cunningham (2008) and Kwok et al. (2009).

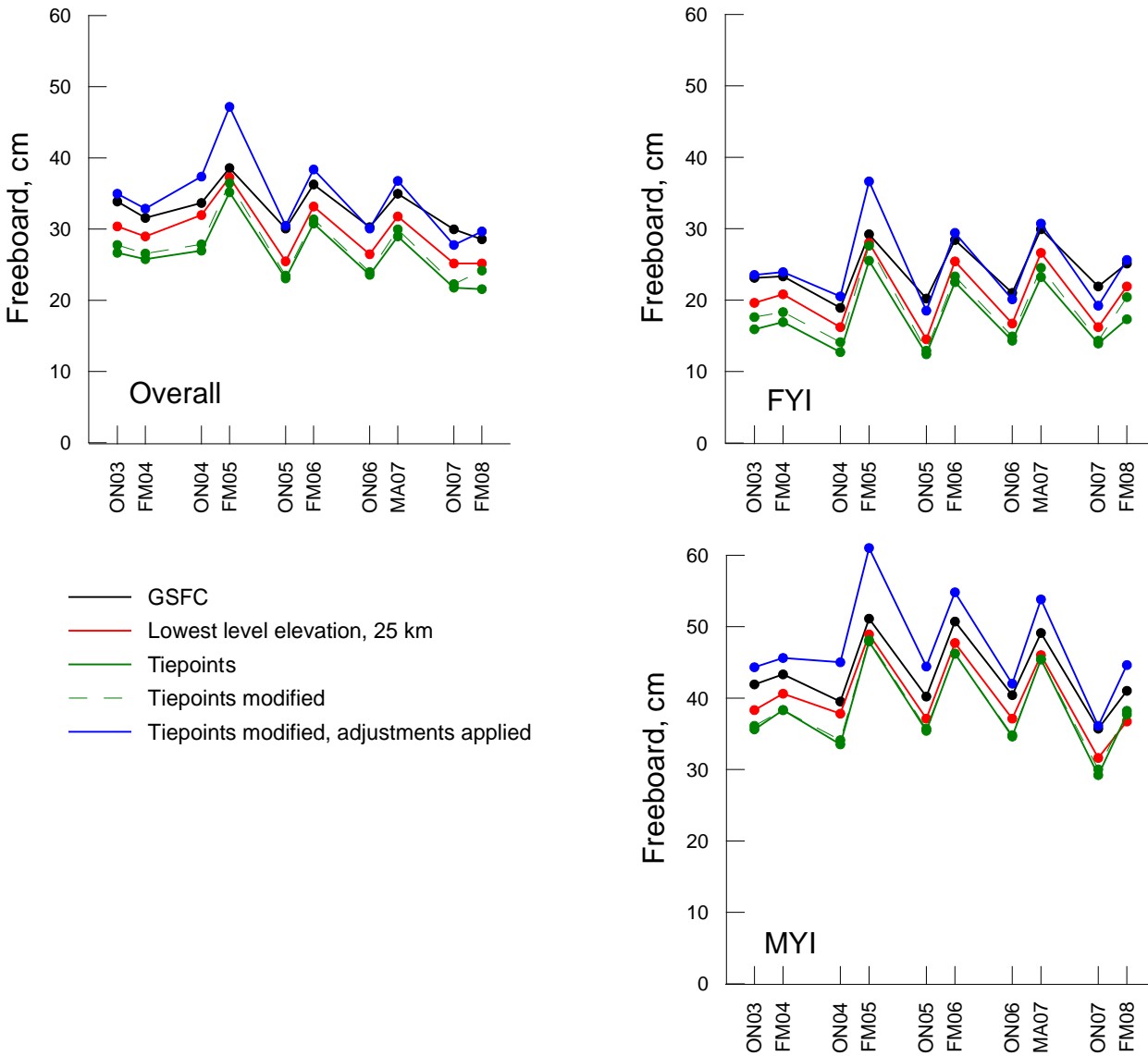

Figure 10. Freeboard time series calculated over the whole arctic basin, FYI and MYI areas as provided in the GSFC product (black) and retrieved in this study by the LLE method when using shorter along-track averaging scales (red), the TP method (green), the TP method, which includes the improvements in the freeboard retrieval algorithm proposed in this study without (green dashed) or with (blue) adjustments for snow depth and lead width applied.

