# Peer review of "On retrieving sea ice freeboard from ICESat laser altimeter"

_The Cryosphere, 2016_

## Referee Comment (RC1) · S. Kern (Referee) · 9 May 2016

Summary: Basically two methods exist to retrieve total (sea ice + snow) freeboard and subsequently

sea ice thickness from ICESat laser altimeter data in the Arctic Ocean. The overall

difference in sea ice thickness between the two methods used, the so-called lowest-levelelevation (LLE) and the tie point (TP) method, has been estimated as 0.42 m. The main aim of

the present paper is to figure out whether this bias is based on the different freeboard

retrieval methods. For this purpose the authors re-construct the methods and investi-

gate the

retrieved freeboard heights as a function of different geoid models and as a function of

different settings required for the two methods used. These settings basically determine how

well the sea surface height (SSH) is approximated against which the freeboard is referenced.

The authors demonstrate that shorter averaging scales are beneficial. They further quantify

the difference in the two approaches due to the different settings and conclude that the TP

method potentially is the one suited better. The authors identify, however, that for thin

sea ice the linear relationship between surface roughness and freeboard, which is used in

the TP method to approximate the SSH, breaks down below a certain freeboard value. The

authors then develop a method how to reduce this bias and quantify the effect of this

improvement, which is valid for first-year ice.

This is an interesting and well-written paper which - even though ICESat is not in orbit

anymore - helps to better understand the limitations involved in sea ice thickness retrieval

using this sensor. The authors managed to convincingly demonstrate that another correction,

when applied to the ICESat freeboard retrieval using the so-called Tie point method,

potentially improves freeboard and hence thickness retrieval even more - and particu-
larly

for first-year ice.

To fully benefit from this improvement and to fully understand the relevance of this

improvement, I feel the paper would benefit from a number of clarifications which are

detailed in the general comments and in the specific comments. In particular, a more
clear

and more focussed description of the goals of the paper, and a revision of the first-year

ice versus multiyear ice discrimination seems to be beneficial for the paper. I therefore

recommend to give the authors the chance for major revisions and optimize the paper
further.

General comments: 1) I have the feeling that the authors could work on the motivation
and the principal aim of

the paper. What are the new findings of this paper? How relevant are these? How
globally

applicable are these? Since ICESat-1 is not in orbit anymore one could ask whether it
is

worth to apply the new approach presented and what a potential user would gain from
that. To

me it seems as if the main improvement is 1-2 cm smaller bias for basically first-year
ice

(FYI). This might be not enough to trigger a potential user to switch to the new, optimal

approach. One could, however, use this paper to give another, additional evidence for

how

difficult it is to derive sea ice thickness from satellite laser altimetry.

2) A relatively large number of the results roots on the QuikSCAT data based discrimination
between FYI and multiyear ice (MYI). The authors used a 50% MYI (or FYI) isoline to separate
between both ice types. I have the feeling that the authors could enhance their results by
reducing the number of mixed pixels and instead of using 50% defining FYI and MYI areas as
follows: FYI: all grid cells with MYI fraction below 10%; MYI: all grid cells with MYI
fraction above 90%. I could imagine that the difference in the freeboard and in the effect
the authors are describing becomes even more clear in that case. And in addition the authors
would concentrate their ice-type based analysis on grid cells where the dicrimination into
these surface types is more reliable and the relative error contribution is smaller than
when using a 50% threshold. Yes, by this the sum of the grid cells analysed (FYI plus MYI)
would not be the total anymore, but I don't think that this would have a significant
influence on results and interpretation.

3) The authors test two different, constant gain settings. I am wondering whether they

are

aware of the paper by Yi et al., 2011, Annals of Glaciology, where it is stated that due to

the instability in the power emitted by the GLAS a constant gain setting might not be

appropriate. Yi et al. (2011) suggest to use a variable gain setting - which was also used

by Kern and Spreen, 2015, for instance. Donghui Yi was one of the main principal scientists

working with ICESat data in the background and I doubt that there are many more who have

more experience with the nitty-gritty technical details of this sensor than Donghui.

4) All maps seem to be blurry compared to the histograms. I am wondering whether the authors

could increase the dpi for these maps. In addition, the color legends in all maps would

benefit from a title and a unit.

Specific comments: Page 1, Line 13: "by up to 15 cm". In the next line the authors give a mean and an "up to"

value. Therefore I would find it fair to see a mean value here as well.

Page 1, Line 15: Perhaps add "average" before "difference"?

Page 1, Line 19: I would not write "is very large and". Here it comes to the main message

the authors wish to give to the readership, i.e. whether the authors wish to give a list of

potential uncertainties and biases or whether the authors wish to promote their improvement.

Page 1, Line 23: One could argue that this statement is not so new but I am wondering

whether authors could take a look the parameters used for the freeboard-to-thickness

conversion. Doesn't the JPL approach use a specifically designed ECMWF data based
snow depth

data set?

Page 2, Line 2: I am wondering whether the authors would like to weaken that "ten
times" by

an "approximately". Currently this contradicts with the notion given on Page 12, lines
30-

32.

Page 2, Line 1-5: Even though the authors refer to the Zygmuntowska et al (2014)
paper later

I suggest that it could be mentioned here as well.

Page 2, Line 8: I am aware of this 2 cm, but wasn't this obtained for a very ideal case
of

sea ice (smooth, extremely level young ice in the Ross Ice Shelf polynya)? I am won-
dering

whether the authors also would like to mention the single-shot accuracy of 13.8 cm.

Page 2, Line 27: "their mean represents" I don't think this is entirely correct. If within a

ground track segment of 550 km the LLE provides 3 elevations then these are used to

approximate the SSH ... maybe by simply computing a mean in the Yi and Zwally
(2009) version

of the LLE but not in the version proposed by Spreen et al. (2006) and Kern and Spreen,

2015, where the SSH at that location is approximated by fitting a polygon through the

elevations identified as minima.

Page 2, Lines 28/29: This statement is true for the missing leads but if the LLE is set to,

e.g. 2% and there are fewer elevations than a certain predefined number (I guess 3 in Kern

and Spreen, 2015; how is this solved in Yi and Zwally?), then for that specific laser shot

no SSH is approximated and it will be instead estimated via interpolation from the

neighboring estimates. The fact that a moving ground track segment is used here ensures that

inconsistencies / jumps are smoothed along track.

Page 2, Line 34: "particular to the chosen value for ice density": Is this true? I thought

that Kwok (et al.), I guess in their 2007 paper, carried out a sensitivity / uncertainty

analysis and figured out that in fact the snow depth is the second most important error

contribution, after the uncertainties in freeboard.

Page 3, Line 10: "is also quantified": I am wondering whether the authors could state which

version of the JPL prodcuts they reproduce: the simple first version or the optimized one.

It is not clear from this paragraph.

Page 3, Line 13: I suggest to add "elevation" behind "level 2"

Page 3, Lines 13-15: Please explain the meaning of "FM" and "MA" as well.

Page 3, Line 16: Is the Yi and Zwally data set also a level 2 data set? The authors could

add this information.

Page 3, Lines 18-21: Please note on which grid projection these two data sets are given.

While the NSIDC data set is publicly available - and may even have a DOI? - the one based on

QuikSCAT seems to be an in-house (NERSC) product. Is this correct? Would it be possible to

give more information about this data set? At the end this is relatively cruicial for your

work because a lot of the results you give are based on the discrimination of FYI and MYI.

Page 3, Lines 24/25: This sentence is not clear to me.

Page 4, Line 2: See general comment 3

Page 4, Line 12: I suggest to write: "... level in leads we evaluate the effect of using a

different geoid on the results in section 3.2" instead of using passive voice.

Page 4, Line 17/18: I sugget to write: "We evaluate and discuss the effect of ..." instead

of using passive voice.

Page 4, Lines 16 & 25: I am wondering whether the authors would consider to add a table in

which they list the different approaches, different scales and different settings used. I can imagine that this would increase the readibility of their paper.

Page 4, Line 24: I suggest to write: "... availability. The distance between ICESat along track samples is 172 m. If we ..."

Page 4, Line 25: It is just a minor detail but I suggest to write "about 580" instead of "about 600".

Page 4, Line 29: I suggest to give more information about the type of satellite images used

by Kwok et al.

Page 5, Line 5: What is R overbar?

Page 5, Lines 9-11: I have difficulties to understand this sentence. What is meant with "their agreement"?

Page 5, Lines 11-12: I have difficulties to understand this sentence as well. Why is the number of tie points not sufficient for basin wide studies?

Page 5, Lines 14-15: I suggest to write: " ... the surface roughness of a given sample ..."

Page 5, Line 15: What is meant by "to be for this sample"?

Page 5, Lines 18-19: I suggest to write: "We discuss the influence of the regression model

in Section 3.3"

Page 6, Lines 1-3: I am aware of this correction. In the frame-work of this paper and these

studies, I suggest the authors could spend 1-2 sentences more on this issue. The TP method

is the one you are finally after here. Now, if the TP method has - as one of the caveats -

that it identifies snow covered, refrozen leads as potential tiepoints, then one could doubt

its reliability. Thin ice, to carry snow on it, needs to have a certain thickness. Could it

be that this correction is also done to take the effect of frostflowers growing on thin ice

into account. These increase the reflectivity as well. In this context the authors could include also a sentence or two into the discussion because

this is an obvious limitation of the TP - or in general of methods using laser altimetry

where the surface reflectivity of the sea ice plays a major role in the SSH approximation.

Page 6, Lines 7-8: I suggest to formulate this more clearly in the last sentence of this

paragraph. The correction does not increase directly with freeboard height because the

latter is not part of the correction factor. Therefore it might be better to write about a

relative increase. In contrast, the reflectivity R is part of the correction factor and

directly influences it.

Page 6, Line 29: Comments to Table 1: Does this table include values from the entire maps

shown in the various figures or did the authors focussed on a certain region? I am asking

because the work of Kwok et al. usually focusses on the Arctic Ocean, omitting seas

such as

the Hudson Bay, Greenland Sea, etc. I am wondering whether it would make sense to do the

same in the present study - and if not what could be a good reason to not limit the area. If

the authors keep the region like they did, then I suggest that they mention this in the

discussion of the results when they are referring to differences between the different

methods in terms of the retrieved sea ice thickness.

Page 6, Line 31: "differences between data releases" I suggest the authors try to be less

global here and to give an approximate estimate of a) what causes these differences and b)

how large these are. This might also help to explain why the standard deviations of line 1

in Table 1 are relatively large.

Page 7, Line 1: "up to 10 cm" Just a comment: This is indeed a lot for sea ice thinner than

15 cm. This is a bias of up to 2/3 of the sea ice thickness and in terms of sea ice

thickness derived from the biased freeboard this would amount to an up to 300% relative

error. That is something.

Page 7, Lines 2-4: See my general comment 3.

Page 7, Line 13: "This explains ..." It seems a bit strange that the usage of a constant

(possibly wrong) gain is selectively causing biases for specific ICESat periods. Again see

general comment 3.

Page 7, Line 24: The authors could add that the main purpose of this along-track averaging

is to get rid of the large-scale fluctuations in the elevations caused by the geoid used.

Page 7, Line 29: "Using ... into ..." I don't understand the meaning here.

Page 7, Line 30: The authors might want to state why they switched to the EGM08 geoid here.

Page 7, Line 31: "from the use of" ... I suggest to write that the authors have obtained

these; these are not just used, right?

Page 8, Line 1: "Freeboard ..." The authors could note that positive biases also occur along

the coasts - something which also confirms the results of Kern and Spreen (2015).

Page 8, Lines 1-2: "patterns are similar" I don't agree here because, e.g., in the Greenland

Sea, the Fram Strait area and the Kara Sea the pattern differ.

Page 8, Lines 3-6: Why should we note the roughness here?

Page 8, Lines 7-8: The text refers to one map in Figure 2d) but the figure actually shows

two maps.

Page 8, Line 11: The authors could mark the location of these ridges in the respective map.

Page 9, Lines 7-11: Why are the authors doing this? I guess I missed the motivation for

discriminating between FYI and MYI. See also my general comment 2.

Page 9, Lines 19-21: I don't understand this sentence. Basin wide and MYI seem to contract

each other. And if the TP method does not provide (enough) freeboard measurements in a 25 km

grid box (or track segment) then there are no data from the TP method, right, and can

therefore not be compared with data from the other method?!?

Page 9, Line 22: (i) can be formulated more clearly perhaps. Isn't the main reason here that

the LLE identifies re-frozen leads, where the refreezing has happened some time ago and

hence the lead is not covered by thin ice anymore, compared to leads with open water or

young ice?

Page 9, Line 23: What does absence of local detection mean? Aren't SSH values interpolated

across gaps?

Page 9, Lines 25-27: The statement that more leads are found over FYI area than in MYI areas

could perhaps be confirmed from the available literature: Willmes and Heinemann, Remote

Sensing, 2016; Ivanova et al., The Cryosphere, 2016; Röhrs et al. The Cryosphere, 2012;

Brohan and Kaleschke, Remote Sensing, 2014.

Page 10, Line 5: "We checked that our results ..." How did the authors do that? What was the

measure used to do this?

Page 10, Line 6: The regression curves mentioned are not shown, right?

Page 10, Line 7: Why "thick MYI"? Isn't simply "MYI" sufficient here?

Page 10, Line 10: Contributions of tie points with "larger distances" dominate? Why larger?

Page 10, Line 11: I am a bit confused here. I thought that sigma_25 is computed for nonoverlapping 25-km segments in the TP method. Did you change this and now use a running 25 km

segment?

Page 10, Line 12: Why do tie points with lower h_r contribute more than those with larger

h_r. I don't fully understand this.

Page 10, Lines 14-15: The authors write about the missing linear relationship between

sigma_25 and freeboard for freeboard values approaching 0 and refer to Figure 6 a) already.

I suggest the authors spend more effort to first discuss Figure 4 in more detail and then

explain what they do and why, by using Figure 5 and 6. Figure 4 did not yet get too

much

attention. See also my comments to Figure 4.

Page 10, Line 18: "contribute more than the other" Why?

Page 10, Lines 28-29: "For example, differences ..." I don't see this.

Page 11, Line 5: "15 samples" as are shown in Figure 5?

Page 11, Line 12: Why is this? The authors could not yet convince me. The dark blue curves,

for instance, seem to be the least noisy ones, for instance.

Page 11, Lines 13-19: The correction is about 1 cm (1-2 cm for FYI) ... is this worth the

effort? Perhaps a more proper discrimination of MYI and FYI would enhance the results (see

general comment 2).

Page 11, Line 18: "differences remain unchanged" This is not true if one looks at Figure 7.

Page 11, general comment to section 3.3.2: I have the feeling that a better description and

discussion of Figure 4 would form a better basis onto which this section can root. I have

the feeling that this section would be more easy to read once the information is structured

a bit better and in chronological order, and by underlining the contribution of the authors

more.

Page 11, Line 24-25: "rather uniformly distributed" Why is this the case? Does this make

sense in the eyes of the authors?

Page 11, Line 29: "from and after ON05" This could be gain related - see general comment 3.

Page 12, Lines 1-2: I am wondering whether it would make sense to show these values in Table

2 as well ... or in a new Table 3 (see below).

Page 12, Line 3: "and over the ICESat period" What is meant here? I would add an "is" before "ranging".

Page 12, general comment: I am wondering why the authors do not show maps generated with

their own "optimal" approach?

Page 12, Lines 8-13: I would find it helpful to have the colors used in Figure 9 to be

written in parentheses in the text after the respective variable. Does the variation of valid grid cells have an influence on your results?

Page 12, Line 19: "it may underestimate ..." Why? Where did the authors demonstrate this? A

link to the respective figure in this paper would help.

Page 12, Line 21: "30 cm" I would read 35 cm from Figure 9.

Page 12, Line 22-23: Are you referring to Figures 4 and 6 here?

Page 12, Line 23: "have greater weight" Why?

Page 12, Line 26: neighboring to leads ... ICESat data." I don't understand how this

worked

out without additional reference data of freeboard; were airborne data used?

Page 12, Line 26-27: "However, the difference between freeboards retrieved ..." Is this the

desired goal? It sounds like GSFC is the truth against which your method can and is

referenced. Is this the case? What I recommend the authors to do is to also discuss the resuls of using the GSFC approach

(LLE) with 25 km window. Figure 9 illustrates that the "regular" GSFC and the 25 km GSFC

differ quite a bit. The authors also mentioned that presumably by using 25 km windowing one

can get more realistic results than using 100 km windowing. I feel that this part of the

discussion is currently missing in the paper.

Page 12, Line 30: The authors please check whether they are indeed referring to "sea ice

freeboard".

Page 12, Lines 30-32: 7 cm ... 6 ... 0.42 m ... these numbers and/or conclusions are

contradicting the notion of the "ten times" mentioned in the introduction.

Page 13, Line 2: "$\sim$ 1 cm" I am suggesting that the authors add another table in which they

show at one glance the results of this paper in comparison to the other results. I guess it

would increase the readability of the paper and would also help the authors as well as

the

readership to estimate the relevance of the optimizations carried out by the authors.

Page 13, Lines 3-7: These lines read as a repetition of information which is also given in

the conclusions.

Page 13, Line 13: Please mention what the database and region for which this number is valid

for.

Page 13, Lines 28-29: "inadequate weighing" To which method does this refer?

Page 13, Lines 29-32: This reads a bit strange. The authors improve the TP such that it

agrees better with the GSFC_100km - over FYI basically - and then recommend to use the TP.

One could ask why not simply use the GSFC? Perhaps the authors can stress more why GSFC

potentially has more uncertainties / biases than the TP ... and the authors could comment on

GSFC_25 km as well.

Page 17, Table 1: I suggest to use +/- instead of the "/" to separate mean and standard

deviations.

Page 18, Table 2: What is the unit? What is the second number in each cell? How do these

results relate to your correction? What is meant by "adjustment" here - or in other

words:

What is the reference? In the second row of the table I would write: "Lead width"

Page 19, Figure 1: - In the maps in b) and c) there is a line artefact in the Fram Strait? The authors could

comment on this. - The histograms might benefit from a vertical line denoting zero difference.

Page 21, Figure 2: - The maps in d) lack the annotation which geoid is used.

Page 23, Figure 3 - Why is the difference between LLE and TP on average smaller for FM periods than for ON

periods? If the positive elevation bias due to re-frozen leads instead of open leads would

be the reason, then I would assume that it is easier to hit an open lead / lead with just

very thin ice during ON periods than during FM periods. And therefore I would assume that

the effect of leads covered with a finite sea ice thickness causing a bias in the LLE and in

turn causing an notable difference between LLE and TP to be larger during FM than during ON.

But apparently the opposite is the case.

Page 25, Figure 4: - The images lacks annnotations a) and b) - I would find it helpful to see lines of the regression used in the TP for every ICESat

period to understand better the limitations of the TP, to understand whether there are

differences between ON and FM periods and learn, where on average, the linear relationship

starts to fail. One could do it via dashed lines like in Figure 6. - I would be good to understand the motivation why the authors did chose the periods shown

in Figure 6. - How would these plots look when FYI and MYI were discriminated?

Page 26, Figure 5: What is the motivation for showing these periods?

Page 27, Figure 6: - The caption lacks description of what the dashed lines refer to. - For images c) I suggest to use the same scale at the y-axis and to perhaps also give the

total number of valid values used in each image (in c).

Page 31, Figure 9: - The thin dashed lines show the results of your optimization. I am wondering why the

authors are not highlighting their results more - see general comment 1. - I am wondering whether it makes sense to show the "old" original TP version results when

the two corrections mentioned (lead fraction in FoV and snow depth / reflectivity) are

needed to come up with "reasonable" results using the TP? One could, instead of showing

results from the TP without these two corrections show only the final versions of the TP. - The LLE does not require any of these two corrections, right?

Typos: Page 1, Line 11: ICEsat –> ICESat

Page 3, Line 15: denoted –> denote

Page 4, Line 28: accordingly –> according

Page 5, Line 7: Kwok et al., –> Kwok et al.

Page 12, Line 25: "in average" –> "on average"

Page 13, Line 20 & 27: "in average" –> "on average"

Page 14, Lines 3-4: "in average" –> "on average"

Page 14, Line 17: "6(69)" –> "56(69"

Page 14, Line 19: Please add "-Oceans" in the Journal title.

---

## Referee Comment (RC2) · Anonymous Referee #2 · 22 Jun 2016

The paper presents a study to evaluate the two NASA methods (TP and LLE) for retrieving total freeboard and explore the similarities and differences for both, and provide an improvement for the TP method to produce a better freeboard than either TP or LLE. The paper is overall read well, although can be improved since many places lost me completely and I have to stop reading. After a few days, I need to reread it again. This is one reason actually took me so long to read it through. I feel the paper has merit for publication, but the presentation can be improved. Clarifications are needed for the followings. (1) in the abstract, you said "LLE give significantly lower estimate over thick multi-year ice and larger estimates over thin first year ice as compared with the TP". However, from figure 9, it seems in all cases, the freeboard from LLE (Goddard) already higher than from TP (Tiepoints). Please explain. (2) in the abstract, line 20-24, you seem say the LLE and TP methods give similar freeboard estimates (that's why
you said the ice thickness difference might be due to different other parameters used). However, from the comment (1) above, it is clear that freeboard from LLE and TP are not the similar. Please explain this statement or rewrite it.

Page 2, Line 24, references for using the LLE method, there is important reference missed and suggest to add in here. Xie, H., A. Tekeli, S. Ackley, D. Yi, and J. Zwally, 2013. Sea ice thickness estimations from ICESat Altimetry over the Bellingshausen and Amundsen Seas, 2003-2009, Journal of Geophysical Research, doi: 10.1002/jgrc.20179

Page 3, line 1, "JPL estimates" to "JPL estimates of ice thickness"

P4-L20, lowest 1

P5-L2, is very confusing, should remove the 25.

P6-L7, "ice" should be "ocean water"?

P6-L10-19, it seems part of the method and should be move to the method. If you can make a kind of work flow chart, it would be even better.

Section 3.5 summary should be combined into the conclusion section

P14-line 7-8, "we obtain similar. . ..", however, figure 9 shows that they are not similar, although the patterns are similar.

Some table and figure captions include a lot of information but a kind of strange way as compared with normal captions of figures and tables. I hope they can make changes and improve them. so readers can easily understand the table or figure. In their current format, I have difficult to read them. for example, Table 1, Figure 2 ((d) is not mentioned in the caption and no unit for any of them)
* * *

---

## Author Comment (AC1) · 21 Jul 2016

**Response to the reviewers**

**'On retrieving sea ice freeboard from ICESat laser altimeter'**

Kirill Khvorostovsky, Pierre Rampal

Nansen Environmental and Remote Sensing Center, 5006 Bergen, Norway

*Correspondence to*: K. Khvorostovsky (kirill.khvorostovsky@nersc.no)

Dear Referees,

Thank you very much for very careful review of the manuscript and many useful comments helping us to improve our paper.

Please find below the answers to your comments and *the changes we have made in the manuscript*:

**1 Response to Referee #1 (Stefan Kern)**

**General comments**

**I have the feeling that the authors could work on the motivation and the principal aim of the paper. What are the new findings of this paper? How relevant are these? How globally applicable are these? Since ICESat-1 is not in orbit anymore one could ask whether it is worth to apply the new approach presented and what a potential user would gain from that. To me it seems as if the main improvement is 1-2 cm smaller bias for basically first-year ice (FYI). This might be not enough to trigger a potential user to switch to the new, optimal approach. One could, however, use this paper to give another, additional evidence for how difficult it is to derive sea ice thickness from satellite laser altimetry.**

In the revised manuscript we have tried to highlight main findings of the paper and to emphasise that the aim of the paper is to analyse the different sources of the uncertainties in freeboard retrieval using ICESat data. In particular, we rewrote most part of the abstract, changed and extended discussion of the results following, in particular, reviewer's comments, and added a paragraph in the conclusions.

**A relatively large number of the results roots on the QuikSCAT data based discrimination between FYI and multiyear ice (MYI). The authors used a 50% MYI (or FYI) isoline to separate between both ice types. I have the feeling that the authors could enhance their results by reducing the number of mixed pixels and instead of using 50% defining FYI and MYI areas as follows: FYI: all grid cells with MYI fraction below 10%; MYI: all grid cells with MYI fraction above 90%. I could imagine that the difference in the freeboard and in the effect the authors are**

**describing becomes even more clear in that case. And in addition the authors would concentrate their ice-type based analysis on grid cells where the discrimination into these surface types is more reliable and the relative error contribution is smaller than when using a 50% threshold. Yes, by this the sum of the grid cells analysed (FYI plus MYI) would not be the total anymore, but I don't think that this would have a significant influence on results and**

5    **interpretation.**

We have checked how the mean freeboards will be changed for FYI and MYI in case of using 10% and 90% thresholds respectively instead of one threshold of 50%. However using of these new thresholds does not noticeably change the difference between the results before and after applying the improved algorithm. In order to emphasize the importance of proposed improvements in the TP method we additionally note their effect to the FM08 period when significant freeboard

10   differences are observed.

**The authors test two different, constant gain settings. I am wondering whether they are aware of the paper by Yi et al., 2011, Annals of Glaciology, where it is stated that due to the instability in the power emitted by the GLAS a**

15   **constant gain setting might not be appropriate. Yi et al. (2011) suggest to use a variable gain setting - which was also used by Kern and Spreen, 2015, for instance. Donghui Yi was one of the main principal scientists working with ICESat data in the background and I doubt that there are many more who have more experience with the nitty-gritty technical details of this sensor than Donghui.**

Indeed, Yi et al., (2011) suggest using varying gain settings for data filtering. In the revised manuscript we have mentioned it

20   in the discussion of freeboard sensitivity to data filtering in section 3.1. However, both, the GSFC and JPL, products were derived by applying a constant threshold as stated in the descriptions of the freeboard retrieval algorithms (i.e. Yi and Zwally, 2009 and Kwok et al., 2007). Since the goal of the paper is to reproduce and compare freeboard retrieval methods used for generation of these two products, we applied constant settings. In this study we focus on the assessment of the effect of different approaches for determination of the sea level references and on the reasons of the differences between the GSFC

25   and JPL products.

**All maps seem to be blurry compared to the histograms. I am wondering whether the authors could increase the dpi for these maps. In addition, the color legends in all maps would benefit from a title and a unit.**

30   The maps were originally produced as png files and their quality remains fine when inserted in the Word document. But, on the stage of creating the pdf document the maps becomes blurred. We suppose that this problem can be solved by providing the original png files or Word document.

We have added the units to the color bars of the maps.

**Specific comments**

**Page 1, Line 13: "by up to 15 cm". In the next line the authors give a mean and an "up to" value. Therefore I would find it fair to see a mean value here as well.**

We have rewrote most part of the abstract to present our findings in the paper more clear. In the new version of the abstract we do not give a mean values for freeboard differences separately for the FYI and MYI.

**Page 1, Line 15: Perhaps add "average" before "difference"?**

Agree, it has been added.

**Page 1, Line 19: I would not write "is very large and". Here it comes to the main message the authors wish to give to the readership, i.e. whether the authors wish to give a list of potential uncertainties and biases or whether the authors wish to promote their improvement.**

Agree, we have removed these words. In the revised manuscript we also rewrote most part of the abstract to present our results more clear.

**Page 1, Line 23: One could argue that this statement is not so new but I am wondering whether authors could take a look the parameters used for the freeboard-to-thickness conversion. Doesn't the JPL approach use a specifically designed ECMWF data based snow depth data set?**

We do not aware of any published studies that analyze the reasons of the difference between JPL and GSFC products. The question whether it is caused by the difference in the methods of freeboard retrieval or in the parameters used to convert freeboard to thickness or both.

Indeed, the GSFC and JPL products were generated using different approaches to determine snow depth. For the JPL product the ECMWF data were used, while for the GSFC product they used Warren climatology modified by applying empirically defined factors, which depend on ICESat period and freeboard values. However, we believe that the effect of the methods for freeboard to thickness conversion deserve separate analysis, while this study is focused on the discrepancies of freeboard retrieval algorithms.

**Page 2, Line 2: I am wondering whether the authors would like to weaken that "ten times" by an "approximately". Currently this contradicts with the notion given on Page 12, lines 30- 32.**

Agree, it has been corrected.

In addition, the words 'sea ice freeboard' used in the section 3.5 (Page 12, lines 32 of the initial version of the manuscript) have been changed to 'total freeboard'.

**Page 2, Line 1-5: Even though the authors refer to the Zygmuntowska et al (2014) paper later I suggest that it could be mentioned here as well.**

Agree, the reference has been added in this paragraph.

**Page 2, Line 8: I am aware of this 2 cm, but wasn't this obtained for a very ideal case of sea ice (smooth, extremely level young ice in the Ross Ice Shelf polynya)? I am wondering whether the authors also would like to mention the single-shot accuracy of 13.8 cm.**

Agree, this accuracy has been mentioned with corresponding reference.

**Page 2, Line 27: "their mean represents" I don't think this is entirely correct. If within a ground track segment of 550 km the LLE provides 3 elevations then these are used to approximate the SSH ... maybe by simply computing a mean in the Yi and Zwally (2009) version of the LLE but not in the version proposed by Spreen et al. (2006) and Kern and Spreen, 2015, where the SSH at that location is approximated by fitting a polygon through the elevations identified as minima.**

Indeed, since in our study we reproduced the LLE method used by Yi and Zwally (2009) for generation of the GSFC product, we focused on the description of their method, which uses mean of the lowest elevation. However mentioning of the fitting method with corresponding references has been added in this paragraph in the revised manuscript.

**Page 2, Lines 28/29: This statement is true for the missing leads but if the LLE is set to, e.g. 2% and there are fewer elevations than a certain predefined number (I guess 3 in Kern and Spreen, 2015; how is this solved in Yi and Zwally?), then for that specific laser shot no SSH is approximated and it will be instead estimated via interpolation from the neighboring estimates. The fact that a moving ground track segment is used here ensures that inconsistencies / jumps are smoothed along track.**

The LLE method, as applied by Yi and Zwally (2009) for the Arctic sea ice, implies using 1% of lowest elevation within running 100-km segment. Thus, they estimate local sea level for all laser shots (remained after data filtering). Therefore, in case if there are no leads within the along track segment the selected lowest elevations would measure the height of thin ice. This has been stated, in particular, by Zwally et al., (2008), where LLE method was proposed, and in the description of the GSFC product (Yi and Zwally, 2009).

**Page 2, Line 34: "particular to the chosen value for ice density": Is this true? I thought that Kwok (et al.), I guess in their 2007 paper, carried out a sensitivity / uncertainty analysis and figured out that in fact the snow depth is the second most important error contribution, after the uncertainties in freeboard.**

That is correct. We have modified this paragraph as follows: *The Arctic sea ice thickness from two available products derived from ICESat data by Jet Propulsion Laboratory (JPL) using the TP method (http://rkwok.jpl.nasa.gov/icesat) and by*

*Goddard Space Flight Center (GSFC) using the LLE method (Yi and Zwally, 2009) were found to be different by 0.42 m (Lindsay and Schweiger, 2015). This difference can be caused by the different techniques for determining the local sea level in the freeboard retrieval algorithm, or by the different methods in estimating snow depth that is used for calculating ice thickness, i.e. by the uncertainty of the freeboard-to-thickness conversion (Kwok and Cunningham, 2008; Zygmuntovska et al, 2014).*

**Page 3, Line 10: "is also quantified": I am wondering whether the authors could state which version of the JPL products they reproduce: the simple first version or the optimized one. It is not clear from this paragraph.**

We reproduce both versions of JPL products in order to assess, from the one side, the effect of using different methods on

10 determination of sea level references, and, from the other side, differences between the freeboards used in the JPL and GSFC products. This sentence has been changed for clarification: *These two corrections were taken into account in the JPL product. Therefore we quantified their effect on freeboard estimates, and hence on the difference between the corresponding sea ice thickness products.*

15 **Page 3, Line 13: I suggest to add "elevation" behind "level 2"**

We are not sure that it would be correct to add "elevation" behind "level 2". Indeed, elevation is the primary product of altimeter data, but they also contain different signal parameters used for data filtering and correction.

**Page 3, Lines 13-15: Please explain the meaning of "FM" and "MA" as well.**

20 We added a sentence with description of these abbreviations denoting ICESat periods: *The abbreviations ON, FM and MA mean October-November, February-March and March-April respectively, followed by the year (i.e. 2003 to 2008).*

**Page 3, Line 16: Is the Yi and Zwally data set also a level 2 data set? The authors could add this information.**

Level 2 altimeter data contain only elevation measurements with corresponding parameters, while the Yi and Zwally data is

25 the final product with the freeboard and thickness estimates. We added word "*product*" in order to emphasize the level of the Yi and Zwally data.

**Page 3, Lines 18-21: Please note on which grid projection these two data sets are given. While the NSIDC data set is publicly available - and may even have a DOI? - the one based on QuikSCAT seems to be an in-house (NERSC)**

30 **product. Is this correct? Would it be possible to give more information about this data set? At the end this is relatively crucial for your work because a lot of the results you give are based on the discrimination of FYI and MYI.**

For more detailed description of these data this sentence has been changed to: *In addition, we use the Arctic-wide multiyear ice fraction dataset used in Zygmuntowska et al. (2014) that was produced by reprocessing the QuikSCAT satellite scatterometer data. Zygmuntowska et al. (2014) used daily averaged gridded (22.5 km) data of radar backscatter processed*

*by Brigham Young University (ftp://ftp.scp.byu.edu/data/qscat/SigBrw), and converted them into multiyear ice fraction following the method described in Kwok (2004).*

**Page 3, Lines 24/25: This sentence is not clear to me.**

We have reformulated this sentence to make it clearer: *When reproducing the TP and LLE methods, we used the same filtering criteria that are applied in Kwok et al. (2007), in order to compare the algorithms avoiding biases associated with different filtering.*

**Page 4, Line 2: See general comment 3**

As we noted in the reply to the general comment 3 we applied a constant setting following the description of the freeboard retrieval algorithms used for generation of the products.

**Page 4, Line 12: I suggest to write: "... level in leads we evaluate the effect of using a different geoid on the results in section 3.2" instead of using passive voice.**

Agree, the sentence has been changed.

**Page 4, Line 17/18: I suggest to write: "We evaluate and discuss the effect of ..." instead of using passive voice.**

Agree, the sentence has been changed.

**Page 4, Lines 16 & 25: I am wondering whether the authors would consider to add a table in which they list the different approaches, different scales and different settings used. I can imagine that this would increase the readibility of their paper.**

Agree, the table with different scales and settings used for the JPL and GSFC products has been added as Table 1. Please, note that following the comment from the reviewer #2 we have also added a flow chart (as the Figure 1) with the methods and settings implemented in our study.

Please, note that due to adding of new Table 1 and Figure 1 in the revised manuscript the numbering of figures and tables is shifted.

**Page 4, Line 24: I suggest to write: "... availability. The distance between ICESat along track samples is 172 m. If we ..."**

Agree, the sentences have been changed.

**Page 4, Line 25: It is just a minor detail but I suggest to write "about 580" instead of "about 600".**

Agree, the number has been changed.

**Page 4, Line 29: I suggest to give more information about the type of satellite images used by Kwok et al.**

We have changed the beginning of the sentence to specify the type of images: *From the analysis of SAR images from RADARSAT satellite, Kwok et al. (2007) found...*

**Page 5, Line 5: What is R overbar?**

In the end of the sentence a definition of $\bar{R}$ has been added: *... with $\bar{R}$ being the mean reflectivity of all the measurements within 25 km around that given sample.*

10 **Page 5, Lines 9-11: I have difficulties to understand this sentence. What is meant with "their agreement"?**

The phrase "their agreement" means agreements of "these two sets of tie-points" mentioned in the beginning of the sentence. We have tried to reformulate this sentence to: *They found good agreement between two sets of tie-points as well as agreement of these sets with high-quality tie-points determined from collocation with satellite images.*

15 **Page 5, Lines 11-12: I have difficulties to understand this sentence as well. Why is the number of tie points not sufficient for basin wide studies?**

The tie-points for the first set are identified as samples with $\Delta R > 0.3$ and located below the regression line. This additional requirements of the dip in reflectivity limits the number of identified tie-points and their sampling density. We have slightly changed the sentence to note it: *Since a sampling density of tie-points in the former set is not sufficient for basin-wide studies*

20 *...*

**Page 5, Lines 14-15: I suggest to write: " ... the surface roughness of a given sample ..."**

Agree, the sentence has been changed.

25 **Page 5, Line 15: What is meant by "to be for this sample"?**

We changed this sentence to make it clearer: *The higher $\sigma_{25}$, which is characteristic of the surface roughness of a given sample, the lower is the $h_r$ is required to qualify this sample as a tie-point.*

**Page 5, Lines 18-19: I suggest to write: "We discuss the influence of the regression model in Section 3.3"**

30 Agree, the sentence has been changed.

**Page 6, Lines 1-3: I am aware of this correction. In the frame-work of this paper and these studies, I suggest the authors could spend 1-2 sentences more on this issue. The TP method is the one you are finally after here. Now, if the TP method has - as one of the caveats - that it identifies snow covered, refrozen leads as potential tiepoints, then one**

**could doubt its reliability. Thin ice, to carry snow on it, needs to have a certain thickness. Could it be that this correction is also done to take the effect of frostflowers growing on thin ice into account. These increase the reflectivity as well. In this context the authors could include also a sentence or two into the discussion because this is an obvious limitation of the TP - or in general of methods using laser altimetry where the surface reflectivity of the sea ice plays a major role in the SSH approximation.**

Two sentences about limitation of the function used to determine the correction have been added: *It can be noted that increase of reflectivity over young ice may also reflect the effect of frost flowers growth. Therefore, the function of reflectivity that account for accumulated snow and used to determine the correction may be different in the presence of frost flowers.*

**Page 6, Lines 7-8: I suggest to formulate this more clearly in the last sentence of this paragraph. The correction does not increase directly with freeboard height because the latter is not part of the correction factor. Therefore it might be better to write about a relative increase. In contrast, the reflectivity R is part of the correction factor and directly influences it.**

The correction does increase directly with freeboard. As we describe in the text, Kwok et al. (2009) calculated the correction as a product of freeboard and a factor $1.1 + 0.1 \left( \frac{R_{snow} - R}{R_{ice}} \right)$. Thus, the first term of this factor (i.e. 1.1) implies that this correction increases all freeboards by at least 10% of freeboard value.

**Page 6, Line 29: Comments to Table 1: Does this table include values from the entire maps shown in the various figures or did the authors focussed on a certain region? I am asking because the work of Kwok et al. usually focusses on the Arctic Ocean, omitting seas such as the Hudson Bay, Greenland Sea, etc. I am wondering whether it would make sense to do the same in the present study - and if not what could be a good reason to not limit the area. If the authors keep the region like they did, then I suggest that they mention this in the discussion of the results when they are referring to differences between the different methods in terms of the retrieved sea ice thickness.**

Indeed, the JPL product does not contains estimates over the seas surrounding the Arctic Ocean (in addition to Hudson Bay and Greenland Sea these are Barents Sea, Kara Sea and Baffin Bay). We believe that it is reasonable not to exclude these areas from the consideration in the sections where we analyze spatial distribution of the difference between the freeboards. Accordingly, we think that it is better to present the average values corresponding to the entire maps in the tables with the differences. However, in the new table proposed below by the reviewer (Table 4 in the revised manuscript) we present the average freeboards that correspond to the areas included in the JPL product. This makes sense because in the section 3.5 we discuss the differences in terms of their effect on the retrieved sea ice thickness and the difference of 0.42 m between the JPL and GSFC products obtained by Lindsay and Schweiger (2015), which corresponds to the Arctic Basin. For consistency, we use corresponding values also in the Figure 9 updated accordingly (Figure 10 in the revised manuscript).

**Page 6, Line 31: "differences between data releases" I suggest the authors try to be less global here and to give an approximate estimate of a) what causes these differences and b) how large these are. This might also help to explain why the standard deviations of line 1 in Table 1 are relatively large.**

5    We suppose that different data filtering is the main factor that causes the freeboard differences in the line 1 in Table 1 (Table 2 in the revised manuscript), and discuss it in more details in the section 3.1. Furthermore, the data from previous releases are not available anymore and it is not possible to provide detailed analysis of the differences between data releases. However, from the description of previous data releases one can presume that one of the most valuable improvements that has been made regards to saturation correction. Therefore, we mentioned this factor in the changed sentence and emphasized

10   that the differences between data releases is secondary factor: *The remaining discrepancies can be attributed to different data filtering, and possibly to the differences existing between data releases (e.g. improvements in saturation correction).*

**Page 7, Line 1: "up to 10 cm" Just a comment: This is indeed a lot for sea ice thinner than 15 cm. This is a bias of up to 2/3 of the sea ice thickness and in terms of sea ice thickness derived from the biased freeboard this would amount**

15   **to an up to 300% relative error. That is something.**

Yes, indeed. This illustrate how the choice of filter settings contribute to uncertainty of the results.

**Page 7, Lines 2-4: See my general comment 3.**

As we noted, in this study we reproduce and compare freeboard retrieval methods used for generation of the GSFC and JPL

20   products, which are derived by applying a constant threshold for the receiver gain parameter. In order to point out to an alternative approach for applying gain setting we have added the following sentences in this paragraph: *It should be noted that Yi et al. (2011) used different thresholds for gain in order to account for the reduce of gain with age of the ICESat lasers due to decrease of the transmitted power. However, in this study, we compare freeboard estimates with the GSFC product derived by Yi and Zwally (2009) using constant setting for the gain threshold.*

**Page 7, Line 13: "This explains ..." It seems a bit strange that the usage of a constant (possibly wrong) gain is selectively causing biases for specific ICESat periods. Again see general comment 3.**

Since Yi and Zwally (2009) apply a constant gain threshold for the GSFC product, it cannot be the reason of the observed difference. At the same time, pulse-broadening parameter also may depend on ICESat's laser age. We have added the

30   following sentences for clarification: *The largest biases and their variability observed in the periods ON03 and ON04 correspond to the first operation periods of laser 2 (campaign 2a) and laser 3 (campaign 3a) respectively. Therefore, one may presume that the pulse-broadening parameter applied by Yi and Zwally (2009) is also affected by the instability in the power transmitted by the ICESat's lasers.*

**Page 7, Line 24: The authors could add that the main purpose of this along-track averaging is to get rid of the large-scale fluctuations in the elevations caused by the geoid used.**

The purpose of the along-track averaging is described in the first sentence of section 2.3, where methodology of the methods is presented. However, we agree with the reviewer, and repeated it here in the revised manuscript as it will be in the line with the response to another reviewer's comment below (Page 7, Line 30 in the initial version of the manuscript). The following sentence has been added: *As described in section 2.3 the running mean $\bar{h}$ is estimated to remove the large-scale fluctuations in the elevations caused by the geoid used.*

**Page 7, Line 29: "Using ... into ..." I don't understand the meaning here.**

The sentence has been corrected as follows: *Using the JPL versus GSFC scales when applying the LLE method results in freeboard differences of about 4 cm"*

**Page 7, Line 30: The authors might want to state why they switched to the EGM08 geoid here.**

The following two sentences about selection of the geoid used have been added: *Note that in our analysis of the freeboard retrieval methods we use data from the newer EGM geoid provided with ICESat data in contrast to the ArcGP geoid used by the JPL and GSFC. Although overall effect of geoid selection on freeboard values is small, we observed some improvements after switching to the EGM geoid, which are discussed in this section below.*

**Page 7, Line 31: "from the use of" ... I suggest to write that the authors have obtained these; these are not just used, right?**

The sentence has been changed to: *By applying other combinations of averaging scales we found that these differences depend mainly on $h_{sl}$ value.*

**Page 8, Line 1: "Freeboard ..." The authors could note that positive biases also occur along the coasts - something which also confirms the results of Kern and Spreen (2015).**

In order to mention a bias along the coast the next sentence after this has been changed to: *A positive bias in freeboard estimates when using longer segments for along-track averaging, as well as tendency for enhanced biases along the coast (see Figure 3), are also reported in Kern and Spreen (2015) for the Weddell Sea in Antarctica.*

**Page 8, Lines 1-2: "patterns are similar" I don't agree here because, e.g., in the Greenland Sea, the Fram Strait area and the Kara Sea the pattern differ.**

Agree, the sentence has been changed to: *Freeboard differences due to different averaging scales for calculation of $\bar{h}$ are small and their patterns have the features of those related to geoid uncertainty shown in Figures 3b and 3c.*

**Page 8, Lines 3-6: Why should we note the roughness here?**

We suppose that the roughness increases the uncertainty in determination of sea level references. In order to elucidate it we have changed this sentence to: *Freeboard differences and their variability associated with along-track averaging scales may also be linked with the surface roughness, which increases uncertainty in determination of the sea level references. This is confirmed by looking at the period FM05 when the largest values of $\sigma_{25}$ are observed (Table 2, line 3 and Figure 6a).*

**Page 8, Lines 7-8: The text refers to one map in Figure 2d) but the figure actually shows two maps.**

We have added '*left'* and '*right'* in the text when referring the Figure (Figure 3d in the revised manuscript).

**Page 8, Line 11: The authors could mark the location of these ridges in the respective map.**

We have marked the location of the Gakkel and Lomonosov ridges on Figure 3c using abbreviations 'G' and 'L' respectively, and added their description in the figure caption.

**Page 9, Lines 7-11: Why are the authors doing this? I guess I missed the motivation for discriminating between FYI and MYI. See also my general comment 2.**

In order to explain a rationale for discrimination between FYI and MYI we have change the sentences and their order in this paragraph as follows: *The obtained freeboard differences are small on average, ranging within ±5 cm, while the presence of significant regional discrepancies should be noted. Since the maps in Figure 4 show a clear distinction between the differences over thin and thick ice for some of the ICESat periods, we estimated differences between freeboards separately for the first-year ice (FYI) and multi-year ice (MYI) regions over the same 25-km grid cells.*

**Page 9, Lines 19-21: I don't understand this sentence. Basin wide and MYI seem to contract each other. And if the TP method does not provide (enough) freeboard measurements in a 25 km grid box (or track segment) then there are no data from the TP method, right, and can therefore not be compared with data from the other method?!?**

We mean that the difference between freeboards would be larger for the means calculated from all grid cells, where freeboard estimates are available over large basin-wide areas (but not only from grid cells, where freeboard estimates available for both datasets). We reformulated the sentence as follows: *One can also expect difference between the basin-wide freeboard means in the area of thick MYI to be even larger when the TP method does not detect any tie-points within some grid cells and, hence, does not provide freeboard estimates.*

**Page 9, Line 22: (i) can be formulated more clearly perhaps. Isn't the main reason here that the LLE identifies re-frozen leads, where the refreezing has happened some time ago and hence the lead is not covered by thin ice anymore, compared to leads with open water or young ice?**

We reformulated this phrase as follows: *(i) the lower-biased estimates in the LLE method due to using the measurements over refrozen leads or ice within the 25-km range for the calculation of local sea level references*

**Page 9, Line 23: What does absence of local detection mean? Aren't SSH values interpolated across gaps?**

Kwok et al. (2008, 2009) fill the gaps after conversion of freeboard to sea ice thickness by interpolation of the gridded estimates, while interpolation of the initial freeboard estimates is not applied.

**Page 9, Lines 25-27: The statement that more leads are found over FYI area than in MYI areas could perhaps be confirmed from the available literature: Willmes and Heinemann, Remote Sensing, 2016; Ivanova et al., The Cryosphere, 2016; Röhrs et al. The Cryosphere, 2012; Brohan and Kaleschke, Remote Sensing, 2014.**

We thank reviewer for providing us with the list of relevant references and added them in this sentence.

**Page 10, Line 5: "We checked that our results ..." How did the authors do that? What was the measure used to do this?**

The sentence has been changed to: *We note that visually the curves corresponding to ON05 and FM06 periods are in agreement with those reported in Kwok et al. (2007).*

**Page 10, Line 6: The regression curves mentioned are not shown, right?**

We believe that all 10 regression lines presented on two plots on the figure (Figure 5 in the revised manuscript) will be tangled and not separated from one another. Therefore, we presented only two averaged regression lines: one for autumn periods and one for winter periods. Following the other reviewer's comments below we also elaborated the section 3.3.2 in order to make it better structured and containing additional description of the figures.

**Page 10, Line 7: Why "thick MYI"? Isn't simply "MYI" sufficient here?**

Agree, it has been changed. Please, note that this sentence has been moved to the second paragraph of this section according to the changes we have made to make the section better structured.

**Page 10, Line 10: Contributions of tie points with "larger distances" dominate? Why larger?**

As stated in the previous sentence the sea level reference for each 25-km segment is estimated by averaging the $h_r$ values corresponding to the tie-points, weighted exponentially by the distance from the regression model. This weighting by the distance implies that larger distances will dominate.

**Page 10, Line 11: I am a bit confused here. I thought that sigma_25 is computed for nonoverlapping 25-km segments in the TP method. Did you change this and now use a running 25 km segment?**

The $\sigma_{25}$ is computed in the same way for both the LLE and TP methods, i.e. as a running mean, while sea level references in the TP method are estimated for non-overlapping 25-km segments using selected tie-points.

**Page 10, Line 12: Why do tie points with lower h_r contribute more than those with larger h_r. I don't fully understand this.**

The tie-points with lower $h_r$ (i.e. more negative values) have larger distance from the regression line (for the same $\sigma_{25}$). Since the sea level reference is estimated by averaging the $h_r$ values weighted by the distance from the regression line, the tie-points with lower $h_r$ have a greater contribution.

**Page 10, Lines 14-15: The authors write about the missing linear relationship between sigma_25 and freeboard for freeboard values approaching 0 and refer to Figure 6 a) already. I suggest the authors spend more effort to first discuss Figure 4 in more detail and then explain what they do and why, by using Figure 5 and 6. Figure 4 did not yet get too much attention. See also my comments to Figure 4.**

We have changed and extended a description of the curves on the Figure (Figure 5 in the revised manuscript) as follows:

*However, as shown by the flattening of the curves in Figure 5, the quasi correlation existing between $h_r$ and $\sigma_{25}$ is lost as it can be seen from the flattening of the curves in Figure 5. Although using a cubic polynomial fit of the data as in Kwok et al. (2007) reduces this flattening, the correlation does not hold towards zero $h_r$ for many ICESat periods. As seen from the averaged regression lines on the Figure 5 a deviation from the linear relationship is more pronounced for the winter periods and starts in a freeboard range from –15 cm to –20 cm. However, we think that this flattening of the curves may not represent an actual and physically-based relationship existing between $h_r$ and $\sigma_{25}$. If this is this case, some samples may be unreasonably identified as tie-points in the TP method due to enlarged area below the regression line at $h_r$ close to zero.*

Then, we discuss regression lines focusing on the selected ICESat periods, i.e. FM05 and FM08 (shown on the Figure 7 in the revised manuscript). In the revised manuscript, we note, in particular, the motivation for selection of these two particular periods. The following sentences after description of the Figure 5 have been added or changed: *The effect of the flattening of the curves on the result of the regression model is illustrated on the Figure 7 for two winter periods: FM05, when the highest $\sigma_{25}$ values are seen for $h_r$ close to zero, and FM08, for which the largest discrepancy between LLE and TP results are observed. The computed regression line for the FM05 period reduces the inversed correlation between $h_r$ and $\sigma_{25}$, but still deviates from the linear relationship (see red dashed lines in Figure 7a). In FM08 the curve is less noisy but correlation is even inversed with an inflection point around $h_r = -5$ cm. As a consequence, the samples detected as tie-points that have a value of $h_r$ close to zero (i.e. measurements taken over areas covered by thiner ice) may contribute more than the other, leading to an artificial increase of the reference sea level height over given segments.*

In the end of this paragraph we provide explanation of missing linear relationship between $\sigma_{25}$ and $h_r$: *The fact that the linear relationship between $\sigma_{25}$ and $h_r$ does not hold for thin freeboard can be explained by a lower likelihood for samples with $\Delta R > 0.3$ to represent actual leads. This is illustrated by the increase of the standard deviation of $\sigma_{25}$ and the decrease in number of samples used in evaluating $\sigma_{25}$ for low absolute values of $h_r$ (Figure 7b and 7c, red). Note that this is consistent with the more pronounced flattening obtained for the winter periods, when variability of the surface roughness is larger.*

**Page 10, Line 18: "contribute more than the other" Why?**

In the inversely distributed part of the relationship between $\sigma_{25}$ and $h_r$ the distance of tie-points from the regression line for a given $\sigma_{25}$ becomes larger toward zero $h_r$. The tie-points with larger distance from the regression line have larger contribution.

**Page 10, Lines 28-29: "For example, differences ..." I don't see this.**

We have reformulated some sentences in this paragraph for more detailed explanation. Since in this paragraph we discuss the Figure 6 (as in the revised manuscript), we have moved here also one sentence with the description of the Figure 6b from the beginning of section 3.3.2. Therefore the second part of the paragraph has been changes to: *In general, the number of tie-points and roughness are anti-correlated and their spatial patterns match very well with the pattern of multi-year versus first-year ice as shown in Figure 6 for some selected ICESat periods discussed in the text, i.e. FM05, ON05, FM06 and FM08. The number of detected tie-points within each 25-km non-overlapping segments ranges from a few tie-points (i.e. < 10) over MYI to several tens (i.e. > 25) over FYI (Figure 6b) and shows a significant spatial variation. A surface roughness represented by $\sigma_{25}$ (Figure 6a) typically do not exceed 10-15 cm over FYI, although it may be locally more than 20 cm as for FM05 period, when the largest roughness is observed (Figure 6a). Therefore, the differences of freeboard estimates in FM05 are primarily related to the surface roughness, and to a lesser degree to number of detected tie-points, which is comparatively low. In contrast, the large number of detected tie points plays a key role in FM08, while surface roughness over FYI is low.*

**Page 11, Line 5: "15 samples" as are shown in Figure 5?**

The figure referred by the reviewer (Figure 6 in the revised manuscript) shows the gridded mean number of tie-points within 25-km non-overlapping segments, while in this sentence we discuss number of samples used to form the curves of relationship between $\sigma_{25}$ and $h_r$. These curves are formed for 1-cm bins for $h_r$ and we require that at least 15 samples should be available for each bin.

**Page 11, Line 12: Why is this? The authors could not yet convince me. The dark blue curves, for instance, seem to be the least noisy ones, for instance.**

In the TP method proposed by Kwok et al. (2007) the tie-points are identified as the points located below the regression line of the relationship between $\sigma_{25}$ and $h_r$. As we described in the section 2.3 they found that the area just below this regression line corresponds to the area around the regression line (+/- 1$\sigma$) of the relationship between $\sigma_{25}$ and the freeboard adjacent to leads identified from SAR images and collocated with ICESat data. For $h_r < -15$ black lines are similar to the red ones, i.e. reproduce the approach by Kwok et al. (2007), while the regression lines derived by applying more stringent selection requirement (cyan and blue lines) are shifted downward compared to the red lines. Thus, selection of tie-points using cyan and blue lines will exclude samples from the area between these lines and black line. Indeed, cyan and blue lines are less noisy. However, this is less important factor because the curves are effectively smoothed when approximation by the cubic polynomial fit is used.

In this sentence we have added some clarifications: *From this analysis, using the condition $h_r < \overline{h_{r25}} - 0.5\sigma_{25}$ in this improved TP method appears to be the most appropriate because it corrects the relationships for thin ice and, at the same time, better reproduces the TP algorithm for $h_r < -15$.*

**Page 11, Lines 13-19: The correction is about 1 cm (1-2 cm for FYI) ... is this worth the effort? Perhaps a more proper discrimination of MYI and FYI would enhance the results (see general comment 2).**

As we noted in the reply to the general comment 2 we have tested the suggestion made by the reviewer about discrimination of MYI and FYI using 10% and 90% thresholds, but it does not result in significant enhancement of the results. Since the most remarkable effect from our improvement of the TP method is revealed for the FM08 period, we have added one sentence to emphasis it: The *most remarkable improvement is observed for the FM08 period when the difference of 5-10 cm over vast areas of FYI are reduced to differences ranging within ±2 cm.*

**Page 11, Line 18: "differences remain unchanged" This is not true if one looks at Figure 7.**

Indeed, for MYI the difference are slightly change, but not significantly. We have corrected the sentence to: *As expected, the differences remain almost unchanged for MYI since our modification of the TP method primarily impacts freeboard estimate over thin ice areas.*

**Page 11, general comment to section 3.3.2: I have the feeling that a better description and discussion of Figure 4 would form a better basis onto which this section can root. I have the feeling that this section would be more easy to read once the information is structured a bit better and in chronological order, and by underlining the contribution of the authors more.**

As noted in our replies to the reviewer's comments above (related to section 3.3.2) and below (related to the figures) we have changed and extended description of the Figures 5, 6 and 7 (as in the revised manuscript), and changed the location of some sentences within the section in order to make it clearer and better structured.

**Page 11, Line 24-25: "rather uniformly distributed" Why is this the case? Does this make sense in the eyes of the authors?**

A correction for snow depth accumulated in leads as estimated using the function proposed by Kwok and Cunningham (2008) is limited by 5 cm. Due to this limit the correction looks uniformly distributed when using the color scale with the range of +/-15 cm, which we use in the manuscript for the maps of freeboard differences. However closer inspection of the maps shows that there are some spatial variations of correction. Therefore, we have changed and extended the description of correction for snow depth as follows: *The adjustment of freeboard included in the TP algorithm and related to snow depth in refrozen leads is limited by 5 cm and is about +2-3 cm on average for all periods considered (Table 3, first line). Due to the fixed limit, this correction is rather uniformly distributed over the Arctic although we note that the lowest values are observed for thin ice in the Arctic seas, i.e. in the warmer regions with slower initial ice growth in leads (Figure 9a for ICESat periods ON05 and FM06).*

**Page 11, Line 29: "from and after ON05" This could be gain related - see general comment 3.**

We believe that correction for lead width being estimated as a product of freeboards and factor of $1.1 + 0.1 \left( \frac{R_{snow} - R}{R_{ice}} \right)$ following Kwok et al. (2009), depends primarily on the freeboard values. This is also confirmed by spatial distribution of the correction, which is correlated with freeboard distribution.

**Page 12, Lines 1-2: I am wondering whether it would make sense to show these values in Table 2 as well ... or in a new Table 3 (see below).**

In the revised manuscript we have added a new table (Table 4 in the revised manuscript) where FYI, MYI and overall results are presented following the reviewer's comment below.

**Page 12, Line 3: "and over the ICESat period" What is meant here? I would add an "is" before "ranging".**

Agree, it has been corrected.

**Page 12, general comment: I am wondering why the authors do not show maps generated with their own "optimal" approach?**

We agree that our own estimates also can be presented. However, as we focus on the analysis of the algorithm performance and their comparison, we suggest to give these freeboard maps in the supplementary materials.

**Page 12, Lines 8-13: I would find it helpful to have the colors used in Figure 9 to be written in parentheses in the text after the respective variable. Does the variation of valid grid cells have an influence on your results?**

Agree, the colors have been written in parentheses in the text.

Selection of the grids available for all considered datasets change the mean values within a range of few centimeters. However, as we noted above, in Table 4 and Figure 10 (as in the revised manuscript) we present the mean values calculated from all grids (i.e. without selection of the collocated grids), but only for the area considered in the JPL product. Therefore, the last sentence in this paragraph have been changed to: *Note that these results correspond to the area considered in the JPL product, i.e. to the Arctic Ocean without surrounding Arctic seas such as Greenland Sea, Barents Sea, Kara Sea and Baffin Bay. Correspondingly, the difference of 0.42 m between sea ice thickness in the JPL and GSFC products found by Lindsay and Schweiger (2015) was derived using randomly selected samples over Arctic Basin. Although excluding of the above-mentioned seas does not significantly impacts the difference between the results, the freeboard means are changed within the range of ±2 cm depending of the ICESat period. The average freeboards obtained from the different methods and corresponding to the results shown on Figure 10 are recapped in Table 4.*

**Page 12, Line 19: "it may underestimate ..." Why? Where did the authors demonstrate this? A link to the respective figure in this paper would help.**

In the revised manuscript we have changed and added the description of the results. In particular, the description of comparison between the freeboards obtained from the TP and LLE method has been extended in the section 3.3.1 and 3.5. Please, find this description below in the response to another reviewer's comment of Page 13, Line 29-32 of the initial version of the manuscript.

**Page 12, Line 21: "30 cm" I would read 35 cm from Figure 9.**

Figure 10 (as in the revised manuscript) shows average freeboard over the whole Arctic, FYI and MYI, while in this sentence we refer only to the lowest part of MYI with freeboard of around 30 cm. In the revised manuscript we rewrote the second and third paragraphs of section 3.5 including the sentence referred by reviewer. This sentence has been changed to: *The lower freeboard obtained by the TP method is mostly observed over FYI and part of MYI in the central Arctic especially in the first three ICESat periods (ON03, FM04 and ON04) (Figures 4 and 10).*

With regards to this, we have also changed one sentence in section 3.3.1 (first sentence in the third paragraph) to: *Positive differences are obtained over large areas of FYI and thin part of MYI for most of the ICESat campaigns, with a peak in FM08.*

**Page 12, Line 22-23: Are you referring to Figures 4 and 6 here?**

That is correct. The references to these figures (5 and 7 in the revised manuscript) have been added in this sentence.

**Page 12, Line 23: "have greater weight" Why?**

As discussed in the sections 2.3 and 3.3.2 sea level references are estimated by averaging the $h_r$ values corresponding to the tie-points and weighted exponentially by the distance from the regression line of the relationship between $h_r$ and $\sigma_{25}$. Tiepoints with lower elevations (more negative $h_r$) have larger distance to the regression line for a given $\sigma_{25}$ and, therefore, have greater weight.

**Page 12, Line 26: neighboring to leads ... ICESat data." I don't understand how this worked out without additional reference data of freeboard; were airborne data used?**

As we describe in section 2.3, Kwok et al. (2007) identified leads using SAR images and then looked on the elevations derived from collocated ICESat measurements. Please, note that in the revised manuscript we split this sentence in to two and moved to the last paragraph of section 3.3.1 where we discuss the reasons of the differences between TP and LLE results. Please, find this description below in the response to the reviewer's comment of Page 13, Line 29-32 in the initial version of the manuscript.

**Page 12, Line 26-27: "However, the difference between freeboards retrieved ..." Is this the desired goal? It sounds like GSFC is the truth against which your method can and is referenced. Is this the case? What I recommend the authors to do is to also discuss the results of using the GSFC approach (LLE) with 25 km window. Figure 9 illustrates that the "regular" GSFC and the 25 km GSFC differ quite a bit. The authors also mentioned that presumably by using 25 km windowing one can get more realistic results than using 100 km windowing. I feel that this part of the discussion is currently missing in the paper.**

We have extended discussion of the differences between the results derived from the LLE method with 25 km and 100 km windows in sections 3.2 and 3.5. The following changes have been made.

In second paragraph of section 3.2 we extended a sentence explaining the difference between freeboards to: *This is expected from the fact that considering a larger window increases the chance to include lower $h_r$ dips in the calculation of $h_{sl}$, although we use the same fraction of the lowest elevations (1%) for selection of tie-points.*

In fourth paragraph of section 3.2 we have changed and extended motivation for choosing of 25-km scale for comparison of the TP and LLE methods by adding one argument, and mention also a drawback of this shorter scale:

*In order to avoid this bias when comparing the LLE and TP methods, we chose to use an averaging window of 25-km to calculate $\bar{h}$ and $h_{sl}$ (as in Kwok et al., 2007) for the three following reasons. First, applying of the TP method using the same scales as in (Kwok et al., 2007) allow us to analyse the performance of the algorithm applied to generate the JPL product. Second, using a smaller window is found to result in reduced dependency of the freeboard on the geoid used in the retrieval process. Indeed, in this case the correction of the geoid for $\bar{h} - h_{sl}$ as well as the fact of using a recent geoid like EGM08 no longer has any impact on freeboard along the ridges in the Arctic, as opposed to what we reported above when using larger spatial averaging. Third reason is that, as demonstrated by Kern and Spreen (2015), a more valid freeboard can be retrieved using the LLE method if the length of along-track segments considered for the selection of the lowest elevations when estimating the local sea level reference $h_{sl}$ is equal to, or less than, the size of the smoothing window used to calculate $\bar{h}$ values. It means that when using 100-km window for determination of $h_{sl}$ the same (or larger) scale would be preferred to*

*calculate $\bar{h}$, so that fluctuations of the geoid would be not properly taken into account properly. From the other side, and as we have shown above, using a shorter length for the averaging windows when applying the LLE method results in lower freeboards that can be interpreted as underestimates due to poorer sampling of the sea level references. The TP methods, in contrast to the LLE method, is based on selection of tie-points using a physical relationship, and its performance can be*

5  *assessed by comparison of the results derived from the TP and LLE approaches.*

In section 3.5 we have extended description of freeboard differences to:

*Because of the use of different averaging scales to calculate sea level references, the sea ice freeboards we estimated using the LLE method with a 25-km averaging window are lower by ~3 cm on average as compared to those of the GSFC product for all ICESat periods (Figure 10, red and black lines). As we discussed in section 3.2, this can be explained by the*

10  *increased likelihood for the lowest elevations to correspond to actual sea level height when using a larger window (e.g. 100 km as applied to produce the GSFC dataset.*

**Page 12, Line 30: The authors please check whether they are indeed referring to "sea ice freeboard".**

Indeed, we refer to the total freeboard, but not sea ice freeboard. This has been corrected.

**Page 12, Lines 30-32: 7 cm ... 6 ... 0.42 m ... these numbers and/or conclusions are contradicting the notion of the "ten times" mentioned in the introduction.**

Indeed, in contrast to the introduction, we refer here to the total freeboard, but not sea ice freeboard. This has been corrected.

20  **Page 13, Line 2: "_ 1 cm" I am suggesting that the authors add another table in which they show at one glance the results of this paper in comparison to the other results. I guess it would increase the readability of the paper and would also help the authors as well as the readership to estimate the relevance of the optimizations carried out by the authors.**

The Table 4 with the mean freeboards corresponding to those presented on Figure 10 as well as the reference to this table in

25  Summary section has been added.

**Page 13, Lines 3-7: These lines read as a repetition of information which is also given in the conclusions.**

Yes, indeed, we repeat the conclusion from the second sentence of this paragraph in the section Conclusions (the first sentence is actually has not been repeated). However, we believe that it is reasonable to repeat one of the main findings of

30  the paper in the Conclusions.

**Page 13, Line 13: Please mention what the database and region for which this number is valid for.**

We added this information in the first paragraph of section 3.5 where we explain that the results presented on the Figure 10 are related to the Arctic Ocean without surrounding seas. Please, find this description above in the response to the reviewer's comment of Page 12, Lines 8-13 of the initial version of the manuscript.

**Page 13, Lines 28-29: "inadequate weighing" To which method does this refer?**

We have specified that this refer to the TP method as follows: ... *their inadequate weighting in these calculations when applying the TP method.*

**Page 13, Lines 29-32: This reads a bit strange. The authors improve the TP such that it agrees better with the GSFC_100km - over FYI basically - and then recommend to use the TP. One could ask why not simply use the GSFC? Perhaps the authors can stress more why GSFC potentially has more uncertainties / biases than the TP ... and the authors could comment on GSFC_25 km as well.**

In sections 3.3.1, 3.4 and 3.5 of the revised manuscript we changed and extended description of the differences between freeboards retrieved by the TP and LLE methods and of the effect of adjustments for snow depth in leads and lead width.

In the last paragraph of section 3.3.1 we added or changed the following sentences:

*The positive differences are most likely reflect the underestimation of freeboard retrieved by the TP method as was found by Kwok et al. (2007) from comparison with the freeboards adjacent to leads detected on satellite images and collocated with ICESat data. They showed that the freeboard underestimation was on average of 1.3 to 4 cm for ON05 and FM06 periods, and explain this by the fact that samples, which are identified as tie-points, do not always represent leads or the thinnest ice. In order to explain why freeboard differences are observed primarily over FYI areas we investigated the performance of the algorithm used in the TP method.*

In the end of section 3.4 we added the following paragraph:

*In principle, the freeboards derived by the LLE method can also be corrected for snow depth in leads and lead width, but this was not done for the GSFC product. Although the LLE method selects only the lowest elevations to determine the local sea level, these samples may be contaminated by snow accumulated in leads or by the neighbouring sea ice surface within the laser altimeter footprint. As shown above from the comparison of freeboards derived by the LLE and TP methods, the LLE method has a weakness mostly over the thickest ice due to lack of leads. The GSFC product, in addition, was derived using longer averaging windows that, in general, increases a likelihood for the lowest elevations to represent a true height of sea level. Therefore, we think that when using the LLE method the empirical functions proposed by Kwok and Cunningham (2008) and Kwok et al. (2009) should be modified depending on the averaging scale applied.*

In section 3.5 we added or changed the following sentences:

*The sea ice freeboards we estimated using the original TP method are lower by ~3 cm on average as compared to those we obtained with the LLE method with identical 25-km along-track averaging scales (Figure 10, green and red lines). As already shown by Kwok et al. (2007), this study shows that the tie-points for determination of sea level references selected by*

*the TP method not always represent leads or the thinnest ice. Although the tie-points with lower elevations have greater weight in most cases, the resulting sea surface reference is biased positive, hence leading to lower freeboard estimates (see Figure 5 and 7). Therefore, we suppose that obtained difference between the TP and LLE results reflects the underestimation of the freeboard derived by the TP method. The lower freeboard obtained by the TP method is mostly observed over FYI and*

5  *part of MYI in the central Arctic especially in the first three ICESat periods (ON03, FM04 and ON04) (Figures 4 and 10). However, the difference between freeboards retrieved by the LLE and TP methods is reduced by more than 30% for the whole Arctic and by 40% for FYI when applying our suggested improvements of the algorithm used in the TP method. At the same time, over thick part of MYI the LLE method tends to give lower freeboards. Although it is not reflected in the mean values on the Figure 10, it can be seen on the maps of the differences (Figure 4 and 8), especially for the FM08 period. This*

10  *is consistent with our expectations that over thick part of continuous MYI with fewer leads, the use of a relationship between the freeboard and surface roughness for identification of tie-points, as done in the TP method, gives more reliable freeboard estimates.*

*… Although the samples used for determination of sea level in the LLE method can also be affected by snow accumulation and contaminated by the neighbouring sea ice surface, these corrections were not applied to freeboards in the GSFC*

15  *product. … This means that, on average, it is sufficient to account for these corrections by using a larger averaging window and LLE method or that both products underestimate freeboard.*

**Page 17, Table 1: I suggest to use +/- instead of the "/" to separate mean and standard deviations.**
Agree, corrected.

**Page 18, Table 2: What is the unit? What is the second number in each cell? How do these results relate to your correction? What is meant by "adjustment" here - or in other words: What is the reference? In the second row of the table I would write: "Lead width"**
We have added the unit (cm) and noted that that second number is standard deviation in the table caption. We use **+/-** instead

25  of the "/" to separate mean and standard deviations as in the Table 1.
We are not sure that we understood reviewer's question: "What is meant by "adjustment" here - or in other words: What is the reference?" In the section 2.5. and 3.4 we describe the meaning of these adjustments. In the table caption we also give the references to Kwok and Cunningham (2008) and Kwok et al., (2009) where these adjustments are proposed.
The title of the second row has been corrected to "Lead width".

**Page 19, Figure 1: - In the maps in b) and c) there is a line artefact in the Fram Strait? The authors could comment on this. - The histograms might benefit from a vertical line denoting zero difference.**
We commented this artefact difference in the figure caption as: *An artefact line of negative differences along the 0 longitude is due to an unexplained positive anomaly in the GSFC freeboard estimates.*

**Page 21, Figure 2: - The maps in d) lack the annotation which geoid is used.**

The annotation for d) has been added.

**Page 23, Figure 3 - Why is the difference between LLE and TP on average smaller for FM periods than for ON periods? If the positive elevation bias due to re-frozen leads instead of open leads would be the reason, then I would assume that it is easier to hit an open lead / lead with just very thin ice during ON periods than during FM periods. And therefore I would assume that the effect of leads covered with a finite sea ice thickness causing a bias in the LLE and in turn causing an notable difference between LLE and TP to be larger during FM than during ON. But apparently the opposite is the case.**

On this figure (Figure 4 in the revised manuscript) the largest negative freeboard differences over MYI are observed for winter periods, e.g. MA07 and FM08. This is exactly because of fewer open leads in winter. For FYI the positive difference are slightly larger for winter periods. As we discuss in section 3.3.2 of the manuscript, this is related to the fact that flattening of the relationship between $\sigma_{25}$ and $h_r$ toward $h_r = 0$ is more pronounced for winter periods due to larger surface roughness.

**Page 25, Figure 4: - The images lacks annnotations a) and b) - I would find it helpful to see lines of the regression used in the TP for every ICESat period to understand better the limitations of the TP, to understand whether there are differences between ON and FM periods and learn, where on average, the linear relationship starts to fail. One could do it via dashed lines like in Figure 6. - I would be good to understand the motivation why the authors did chose the periods shown in Figure 6. - How would these plots look when FYI and MYI were discriminated?**

- The annotations has been added to the figure.

- We have added two average regression lines for autumn and winter periods and discuss them in the section 3.3.2. They show the difference between ON and FM periods and where, on average, the linear relationship starts to fail. A discussion of limitation of the TP method has been extended in the description of the Figures 5 and 7 (as in the revised manuscript), and in the end of section 3.3.1.

- As we noted in the response to the reviewer's comments above we extended and changed description of the Figures 5 and 7 (as in the revised manuscript) in section 3.3.2, where we mentioned the motivation for selection of the periods.

- The measurements over FYI are represented in the part of the curves form 0 to 15-20 cm, while MYI is reflected in the rest part of the curves. Therefore, we believe that curves derived separately from FYI and MYI would be similar to those two parts of the relationship.

**Page 26, Figure 5: What is the motivation for showing these periods?**

These four periods were selected as the periods discussed in the manuscript in more details. We have mentioned it in the second paragraph of the section 3.3.2 where this figure is discussed.

**Page 27, Figure 6: - The caption lacks description of what the dashed lines refer to. -For images c) I suggest to use the same scale at the y-axis and to perhaps also give the total number of valid values used in each image (in c).**

- Annotation for the dashed lines has been added.

- We have also added the total numbers of valid values in c). However, we are not sure that using the same scale at the y-axis is reasonable because in this case it will be hard to see the difference between the curves for the FM08 period.

**Page 31, Figure 9: - The thin dashed lines show the results of your optimization. I am wondering why the authors are not highlighting their results more - see general comment 1. - I am wondering whether it makes sense to show the "old" original TP version results when the two corrections mentioned (lead fraction in FoV and snow depth / reflectivity) are needed to come up with "reasonable" results using the TP? One could, instead of showing results from the TP without these two corrections show only the final versions of the TP. - The LLE does not require any of these two corrections, right?**

- As we noted in the answer to the general comment 1 we tried to highlight our results in the context of the difficulties of deriving sea ice freeboard and thickness from satellite laser altimetry. Also we pointed out significant improvement for the FM08 period.

- In principle, the freeboards derived by the LLE method also can be corrected for snow depth in leads and lead width, though it has not been done in the GSFC product. Although the LLE method selects only the lowest elevations for the tie-points these samples also may be contaminated by snow accumulated in leads or by the neighbouring sea ice surface within the laser altimeter footprint. Therefore, we think that it is reasonable to show the results derived by the LLE and TP methods without these corrections in order to show the difference caused solely by the method of sea level reference determination. We have added discussion of the effect of these two corrections in the revised manuscript. Please, find this description above in the response to the reviewer's comment of Page 13, Line 29-32 in the initial version of the manuscript.

**Typos: Page 1, Line 11: ICEsat –> ICESat**

Agree, it has been corrected.

**Page 3, Line 15: denoted –> denote**

Agree, it has been corrected.

**Page 4, Line 28: accordingly –> according**

Agree, it has been corrected.

**Page 5, Line 7: Kwok et al., –> Kwok et al.**

Agree, it has been corrected.

**Page 12, Line 25: "in average" –> "on average"**

Agree, it has been corrected. Please, note that the revised manuscript this sentence has been moved to the last paragraph of section 3.3.1 where we discuss the reasons of the differences between TP and LLE results.

**Page 13, Line 20 & 27: "in average" –> "on average"**

Agree, it has been corrected.

**Page 14, Lines 3-4: "in average" –> "on average"**

Agree, it has been corrected.

**Page 14, Line 17: "6(69)" –> "56(69"**

Agree, it has been corrected.

**Page 14, Line 19: Please add "-Oceans" in the Journal title.**

We are not sure that this comment is correct. Paper by Kurtz and Markus ("Satellite observations of Antarctic sea ice thickness and volume") is published in the Journal of Geophysical Research, but not Journal of Geophysical Research Ocean.

**2 Response to anonymous referee #2**

**General comments**

**I feel the paper has merit for publication, but the presentation can be improved.**

We have substantially changed and extended description of the results and figures in section 3 in the revised manuscript, in particular, by accommodating the reviewers' comments.

**Specific comments**

**(1) in the abstract, you said "LLE give significantly lower estimate over thick multi-year ice and larger estimates over thin first year ice as compared with the TP". However, from figure 9, it seems in all cases, the freeboard from LLE (Goddard) already higher than from TP (Tiepoints). Please explain.**

In this sentence we note that these differences are derived when applying the same along-track averaging scales, i.e. we refer the results denoted as 'Tiepoints' and 'Lowest level elevation, 25 km' on the Figure 9. The GSFC product is obtained using

longer along-track averaging scale that increases freeboard estimates. We have rewrote most part of the abstract to present our findings in the paper more clear.

**(2) in the abstract, line 20-24, you seem say the LLE and TP methods give similar freeboard estimates (that's why you said the ice thickness difference might be due to different other parameters used). However, from the comment (1) above, it is clear that freeboard from LLE and TP are not the similar. Please explain this statement or rewrite it.**

We found that LLE and TP methods give on average similar freeboard estimates if we apply the settings in the freeboard retrieval algorithms as were used to generate JPL and GSFC products. The algorithms applied to produce these two datasets have three differences: 1) method for determination of the sea level references (LLE and TP methods), 2) along-track averaging scale (longer for GSFC product and shorter for JPL product), 3) corrections for snow depth and lead width was applied only in the JPL product. We assess the effect from all these factors and conclude that on average they should roughly compensate each other. The differences noted in the sentence discussed in the comment (1) describe the effect of the difference '1)', i.e. difference of the method for determination of the sea level references. We have rewrote most part of the abstract to present our findings in the paper more clear.

**Page 2, Line 24, references for using the LLE method, there is important reference missed and suggest to add in here. Xie, H., A. Tekeli, S. Ackley, D. Yi, and J. Zwally, 2013. Sea ice thickness estimations from ICESat Altimetry over the Bellingshausen and Amundsen Seas, 2003-2009, Journal of Geophysical Research, doi:10.1002/jgrc.20179**

In the revised manuscript we have included this reference in the list of relevant papers.

**Page 3, line 1, "JPL estimates" to "JPL estimates of ice thickness"**

We have reformulated this paragraph. The sentence referred by the reviewer has been combined with the first sentence of this paragraph and changed to: T*he Arctic sea ice thickness from two available products derived from ICESat data by Jet Propulsion Laboratory (JPL) using the TP method (http://rkwok.jpl.nasa.gov/icesat) and by Goddard Space Flight Center (GSFC) using the LLE method (Yi and Zwally, 2009) were found to be different by 0.42 m (Lindsay and Schweiger, 2015).*

**P4-L20, lowest 1**

Agree, this phrase has been changed to: … l*owest 1% of the $h_r$ values...*

**P5-L2, is very confusing, should remove the 25.**

We are not sure that removing '25-km' is reasonable here. Kwok et al. (2007) could, in principle, use another window size to estimate $\sigma_{25}$. In addition it explains a subscript for $\sigma_{25}$.

**P6-L7, "ice" should be "ocean water"?**

When defining the variables in this scaling factor, Kwok et al. (2009) describe $R_{ow}$ as reflectivity of bare ice. In order to avoid inconsistency we have changed this subscript to 'ice', i.e. to $R_{ice}$.

**P6-L10-19, it seems part of the method and should be move to the method. If you can make a kind of work flow chart, it would be even better.**

In the section 2 we describe the methods of freeboard retrieval that we reproduce, while in this paragraph we outline the results of freeboard comparison. Therefore, we believe that it is more reasonable to keep this part as the introduction of section with the results.

We have made a flow chart of the methods and settings implemented in our study as a new Figure 1. Please, note that due to adding of new Figure 1 and Table 1 in the revised manuscript the numbering of figures and tables is shifted.

**Section 3.5 summary should be combined into the conclusion section**

We agree that, in principal, this section could be combined with the conclusions. However, the Summary section presents not only overview of the results obtained in this study, but we also discuss the mean time series on the (Figure 10 in the revised manuscript) and mean freeboards in the new Table 4 that have not been presented before. Therefore, we think that this summary is more relevant to the section with the results.

**P14-line 7-8, "we obtain similar: : :.", however, figure 9 shows that they are not similar, although the patterns are similar.**

As we noted in the reply to comments (1) and (2) the results become on average similar when we apply the settings in the freeboard retrieval algorithms as were used to generate JPL and GSFC products. The results on the Figure 10 (as in the revised manuscript) denoted as 'Tiepoints' (green) and 'Tiepoints modified' (green dashed) are obtained without corrections for snow depth in leads and lead width. After applying these corrections the results (blue line) become close to the GSFC product (black).

**Some table and figure captions include a lot of information but a kind of strange way as compared with normal captions of figures and tables. I hope they can make changes and improve them. so readers can easily understand the table or figure. In their current format, I have difficult to read them, for example, Table 1, Figure 2 ((d) is not mentioned in the caption and no unit for any of them)**

The annotation for d) on the Figure 3 (as in the revised manuscript) has been added.

Indeed, the captions to the Table 2 and Figures 2 and 3 (as in the revised manuscript) contain a lot of information. However, on these figures and table we a various sort of information on different plots (in the figures) and lines (in the Table 2). Therefore, it seems difficult to shorten this information without loosing details that are needed to understand their content.

[revised manuscript text omitted]

---

## Referee Report (RR1)

[referee-annotated manuscript omitted]

---

## Author Response (AR2)

**Response to the reviewer (Stefan Kern)**
**'On retrieving sea ice freeboard from ICESat laser altimeter'**

Kirill Khvorostovsky, Pierre Rampal

Nansen Environmental and Remote Sensing Center, 5006 Bergen, Norway

*Correspondence to*: K. Khvorostovsky (kirill.khvorostovsky@nersc.no)

Dear reviewer,

Thank you very much for your editorial suggestions and comments of the revised manuscript, which helped us to improve and finalize the manuscript. Please find below the answers to your comments followed by a corrected manuscript with track changes.

**Dear authors,**

**you did a great job in replying to my comments of the first review and I have no further comments. However, there is a number of editoral suggestions which you will find in the pdf-document of your revised manuscript into which I inserted the comments directly. I urge you to check for the usage of "sea ice freeboard" and "freeboard", and for**

**consistent use of terms such as "regression curve" or "line" and "sea surface reference" versus "sea level reference". The reference list needs a careful check for "." and ",".**

We have made changes in the manuscript following the reviewer's comments and suggestions. In particular, we incorporated reviewer's editorial suggestions, edited some sentences to make them more clear, and checked and corrected the usage of the terms for consistency.

**I have one major comment with the abstract which you will find inserted at the end of the abstract in the pdf document. I have the feeling that the abstract can still be more clear and I hope the authors will take the respective action. Still I put this under "technical corrections" and accept the manuscript as is otherwise.**

We have made changes in the Abstract in order to make it more clear following the comments and suggestions from the reviewer.

[revised manuscript text omitted]